# Proteome Mapping of South African Cassava Mosaic Virus-Infected Susceptible and Tolerant Landraces of Cassava

**DOI:** 10.3390/proteomes9040041

**Published:** 2021-10-23

**Authors:** Elelwani Ramulifho, Marie Emma Christine Rey

**Affiliations:** 1Plant Biotechnology Laboratory, School of Molecular and Cell Biology, University of the Witwatersrand, Johannesburg 2001, South Africa; 488507@students.wits.ac.za; 2Germplasm Development, Agricultural Research Council-Small Grain Institute, Bethlehem 9700, South Africa

**Keywords:** cassava, recovery phenotype, tolerance, *South African cassava mosaic virus*, TME3 landrace, T200 landrace, geminivirus, protein–protein network, chloroplast, ribosome

## Abstract

The production of cassava is threatened by the geminivirus South African cassava mosaic virus (SACMV), which causes cassava mosaic disease. Cassava landrace TME3 shows tolerance to SACMV, while T200 is highly susceptible. This study aimed to identify the leaf proteome involved in anti-viral defence. Liquid chromatography mass spectrometry (LC-MS) identified 2682 (54 differentially expressed) and 2817 (206 differentially expressed) proteins in both landraces at systemic infection (32 days post infection) and symptom recovery (67 days post infection), respectively. Differences in the number of differentially expressed proteins (DEPs) between the two landraces were observed. Gene ontology analysis showed that defence-associated pathways such as the chloroplast, proteasome, and ribosome were overrepresented at 67 days post infection (dpi) in SACMV-tolerant TME3. At 67 dpi, a high percentage (56%) of over-expressed proteins were localized in the chloroplast in TME3 compared to T200 (31% under-expressed), proposing that chloroplast proteins play a role in tolerance in TME3. Ribosomal_L7Ae domain-containing protein (Manes.12G139100) was over-expressed uniquely in TME3 at 67 dpi and interacts with the ribosomal protein Sac52 (RPL10). RPL10 is a known key player in the NIK1-mediated effector triggered immunity (ETI) response to geminivirus infection, indicating a possible role for Sac52 in SACMV recovery in TME3. In conclusion, differential protein expression responses in TME3 and T200 may be key to unravel tolerance to CMD.

## 1. Introduction

Cassava (*Manihot esculenta* Crantz), a perennial shrub and root crop belonging to the *Euphorbiaceae* family, is the third most important food crop in tropical regions after rice and maize. It is grown in over 90 countries as it is a good source of carbohydrates and produces higher yields in adverse conditions compared to most other crops. According to Faostat (http://www.fao.org/faostat/, accessed on 11 August 2021), over 303 million tonnes of cassava were produced in 2019, with Africa being the major contributor (63.3% production share). Cassava does not only provide food but also brings employment to the impoverished regions of Africa. Cassava has many other uses such as in industries producing food, ethanol, paper and cardboard, textiles, pharmaceuticals, and glues and adhesives. Cassava roots are high in carbohydrates, while its leaves are an excellent source of both proteins and vitamins [1]. There is no doubt that the cassava crop is of importance; however, its production is threatened by pests and pathogens, resulting in yield losses.

South African cassava mosaic virus (SACMV) [2] is amongst nine cassava mosaic geminivirus species [3] which cause the devastating cassava mosaic disease (CMD) [4,5]. SACMV has a bipartite single-stranded DNA genome consisting of DNA-A and DNA-B components, both crucial for systemic infection [6]. DNA-A is necessary for transcription and replication of the virus, while DNA-B is crucial for cell-to-cell and long-distance movement [7]. DNA-A encodes the replication-associated protein (Rep) (AC1), transcriptional activator protein (TrAP) (AC2), replication enhancer protein (REn) (AC3), RNA-silencing suppressor (AC4), coat protein (CP) (AV1), and the pre-coat protein (AV2). DNA-B encodes the nuclear-shuttle protein (NSP) (BV1) and the movement protein (MP) (BC1) [8]. In addition to their function in infection, these encoded proteins also influence defence responses, hormonal responses and cell cycle regulation, ubiquitin-proteasomal pathways, and protein-signalling cascades [9]. Since its first report in the late 19th century, CMD has resulted in major cassava production and economic losses to the value of USD 1300–2300 million in Africa [10]. In susceptible cultivars, CMD has caused yield losses up to 82% [11] by significantly reducing the number and size of tubers. However, the severity of the damage is determined by the strain of the virus, cassava variety, age and sensitivity of the host, and environmental factors such as soil fertility [12]. CMD presents itself with symptoms of yellow and/or pale green chlorotic mosaic of leaves accompanied by distortion and an overall reduction in leaf size [13].

Cassava landraces TME3 and T200 have shown different responses to SACMV at different infection stages. TME3 is tolerant to SACMV, as shown by a recovery phenotype approximately 67 days post infection (dpi), whereas T200 is susceptible to the virus throughout the infection period. Findings from a transcriptome profiling study showed many down-regulated and up-regulated genes in the two landraces, including several resistance (R) gene transcripts over-expressed in TME3 and under-expressed in T200 [14]. Although this study revealed some interesting findings which could potentially give some clues to the molecular mechanisms that underly the tolerance phenotype of TME3, levels of mRNA in the cell are not directly proportional to the expression level of their cognate proteins. Proteomics studies the complete set of proteins expressed in a specific cell, tissue, or organism and thus bridges the gap between the other omics [15]. As such, it is important to look at the proteome as this offers a more robust approach to discovering the protein networks and pathways that are crucial for stress responsiveness and disease tolerance, and it will enable further dissecting of the biochemical and molecular responses of cassava against viruses.

The number of proteome mapping studies of crops has increased, and so has the pool of proteomic databases that are now available in the research community. In addition, proteomics has other potentials in the biotechnology field, including the identification of molecular markers to assist in plant breeding, keeping track of changes in protein profiles under different abiotic and biotic treatments at different timelines, understanding pathogen resistance or tolerance mechanisms, and the possibility of developing proteomic-based pesticides [16,17]. Together with the availability of fully sequenced genomes, researchers are now able to integrate different omics data into systems biology in order to seek an understanding of the physiological systems of a cell or tissue in response to different stimuli. Gel-based proteomics techniques such as two-dimensional electrophoresis (2-DE), and gel-free techniques such as mass spectrometry (MS)-based and Isobaric tag for relative and absolute quantitation (iTRAQ), have been used to identify proteomes in different crops [18]. Recently, a review examined tomato proteomes performed on different tissues using different proteomic techniques under different stress/physiological conditions [19]. Two of these studies investigated the differences in protein profiles in tomato fruit infected with tobacco mosaic virus (TMV) [20]. Pathogenesis-related (PR) proteins such as CHI and GLU and antioxidant enzymes such as SOD, DHAR, GST, PHGPX, APX, and TPX were found to protect the plant against TMV infection [20]. PR proteins were further confirmed to be involved in defence against TMV infection in model plant *Nicotiana tabacum* [21]. Global plant proteome changes during viral infection have been reported in several other studies, showing a range of functions impacted such as translation, protein processing and degradation, intracellular trafficking, primary metabolism, amino acid metabolism, stress responsive proteins, and cell wall biogenesis [22]. In tomato plants infected with cucumber mosaic virus (CMV), the responsive leaf proteome revealed proteins involved in photosynthesis, primary metabolism, and defence activity to be under-expressed following infection [23]. Several proteome studies have been performed on geminivirus-tomato interactions, comparing resistant and susceptible germplasm and revealing hypothetical models for host cell responses to infection [24,25]. A cassava proteome has been investigated in storage roots and leaves under abiotic stress and physiological conditions [26,27,28,29,30,31]; however, there has only been one study that investigated the leaf proteome in response to viral infection. In this study, cassava was infected with both Indian cassava mosaic virus (ICMV) and Sri Lankan cassava mosaic virus (SLCMV), which causes CMD [32]. Proteins such as the DnaJ-like proteins involved in molecular chaperone functions, signalling, and disease resistance (ascorbate peroxidase and protein kinase-coding resistance proteins) were over-expressed, implying their involvement in viral infection resistance. Proteomes have also been established from other crops such as sorghum, rice, soybean, wheat, maize, and mungbean [33,34,35,36,37,38] in response to different stimuli.

This study is the first to profile a cassava proteome in response to infection by a geminivirus, SACMV, on the sub-Saharan African continent. Furthermore, the study compares changes in cellular proteins between a susceptible T200 and tolerant TME3 landrace at the systemic infection stage 32 days post inoculation (dpi) and symptom recovery (67 dpi). Overall, the results showed differences in protein expression profiles between the two landraces and also revealed proteins that are proposed key players in reduced symptoms and virus tolerance in TME3. The results will go towards understanding the biochemical mechanisms involved in both susceptibility and symptom/virus recovery phenotypes in cassava infected with CMD.

## 2. Materials and Methods

### 2.1. Cassava Infection with SACMV

Nodal cutting of cassava T200 and TME3 landraces were micro-propagated on full Murashige and Skoog (MS) medium [39] supplemented with 2% sucrose, 0.78% plant tissue culture agar, and 0.002 mM CuSO_4_, with a pH of 5.8. To induce root growth, the nodal cultures were incubated in a temperature-controlled growth room at 28 °C under a 12 h photoperiod at a light intensity of 150 μEm^−2^/s. Once the roots appeared (approximately 3 weeks), the plantlets were transplanted into 40 mm × 45 mm Jiffy^®^ pellets (Jiffy International AS, Kristiansand, Norway), placed in a plastic container, and covered with a plastic film for acclimatization. Two slits were made to the plastic film to gradually acclimatize plantlets until they could be infected with SACMV at a 4–6 leaf stage. Agro-inoculation of the plantlets with SACMV infectious dimer clones, DNA A, and DNA B, in *Agrobacterium* Agl1 was performed as outlined by [14], with minor modifications. *Agrobacterium* harbouring the infectious clones were grown in yeast extract peptone medium (1% tryptone, 1% Bacto peptone, 0.5% sodium chloride, pH 7) supplemented with 50 mg/mL carbenicillin and 100 mg/mL kanamycin, while *Agrobacterium* Agl1 medium was only supplemented with 50 mg/mL carbenicillin. The cultures were grown at 30 °C until an OD600 between 1.8–2.0 was reached. SACMV DNA A and DNA B cultures were mixed in equal volumes before plantlets were agro-inoculated along the stem with 100 μL of the culture using a hypodermic needle. Control plants were mock-inoculated with *Agrobacterium* Agl1 only, incubated in a temperature-controlled growth room, and monitored over a period of 67 days.

### 2.2. Protein Extraction and Isolation by TCA/Acetone Precipitation

Leaves below the apex from SACMV-infected and mock-inoculated plants were sampled at time points 32- and 67-days post inoculation (dpi). Three infection trials (biological replicates) were carried out independently for both T200 and TME3 landraces, with two leaves from a single plant and nine plants sampled per treatment per trial. Leaves from 3 plants were pooled into one tube/sample, and each treatment had 3 samples. Sampled leaves were immediately flash frozen in liquid nitrogen to halt any metabolic activity. A total of 2 g of crushed leaf samples was used for protein extraction using a HEPES-based buffer (50 mM HEPES-KOH, pH 7.4; 100 mM KOAc; 2 mM MgCl2; 0.5 mM PMSF; and 1% Proteinase inhibitor (ThermoFisher Scientific, Waltham, USA, 78438), 0.1% Triton x-100). Samples containing the extraction buffer were gently mixed in a low-speed vortex and placed on ice for 30 min before centrifuging at 4 °C for 10 min at 3000 rpm. A total of 400 μL lysate was transferred to a 2 mL tube. Proteins were precipitated by the addition of 1.6 mL of ice-cold TCA extraction buffer (10% *v*/*v* TCA (trichloroacetic acid; Sigma-Aldrich, Darmstadt, Germany, T9159) in acetone (Sigma-Aldrich, 34850)) with 2% 2-mercaptoethanol (Honeywell, Charlotte, USA, 63700) and incubated at −20 °C for a minimum of 1 h. Protein was pelleted by centrifugation at 21,000× *g* for 15 min at 4 °C. The supernatant was then removed and the pellet washed with 500 μL of ice-cold acetone. This was repeated a further two times for a sum of three washes in total. The pellets were then dried in a fume hood. Protein pellets were solubilised in 20 μL of 4% sodium dodecyl sulphate (SDS; Sigma-Aldrich, 71736) and 100 mM triethylammonium bicarbonate (TEAB, Sigma-Aldrich, T7408), briefly vortexed, and then heated at 95 °C for 5 min. Insoluble debris was removed by centrifugation at 10,000× *g* for 5 min. The supernatant was then transferred to a new tube and quantified using the QuantiPro BCA assay kit (Sigma-Aldrich, QBCA) according to the manufacturer’s instructions.

### 2.3. On-Bead Hydrophilic Interaction Liquid Chromatography (HILIC) Digest

In preparation for the HILIC magnetic bead workflow, the HILIC beads (ReSyn Biosciences, Pretoria, South Africa, HLC010) were aliquoted into a new tube and the shipping solution removed. Beads were then washed with 250 μL wash buffer (15% acetonitrile (ACN), 100 mM ammonium acetate (Sigma-Aldrich, 14267) pH 4.5) for 60 s. This was repeated for a total of two washes. The beads were then resuspended in loading buffer (30% ACN, 200mM ammonium acetate pH 4.5) to a concentration of 5 mg/mL. A total of 50 μg of protein from each sample was transferred to a 96-well protein LoBind plate (Merck, Darmstadt, Germany, 0030504.100). Protein was reduced with tris (2-carboxyethyl) phosphine (TCEP; Sigma-Aldrich, 646547), which was added to a final concentration of 10 mM TCEP and incubated at 60 °C for 1 h. Samples were cooled to room temperature and then alkylated with methyl-methane-thiosulphonate (MMTS; Sigma-Aldrich, 208795), which was added to a final concentration of 10 mM MMTS and incubated at room temperature for 15 min. HILIC magnetic beads were added at an equal volume to that of the sample and a ratio of 5:1 total protein. The plate was then incubated at room temperature on the shaker at 900 RPM for 30 min for binding of protein to beads. After binding, the beads were washed four times with 500 μL of 95% ACN for 60 s. For digestion, trypsin (Promega, Madison, USA, PRV5111), made up in 50 mM TEAB, was added at a ratio of 1:12.5 total protein, and the plate was incubated at 37 °C on the shaker for 4 h. After digestion, the supernatant containing peptides was removed and dried down. Samples were then resuspended in LC loading buffer (0.1% FA, 2% ACN) prior to clean up by Zip-Tip (Sigma-Aldrich, Z720070). Samples were then dried down once more and resuspended in a final volume of 12 μL LC loading buffer.

### 2.4. Liquid Chromatography-Mass Spectrometry (LC-MS)

LC-MS analysis was conducted with a Q-Exactive quadrupole-Orbitrap mass spectrometer (ThermoFisher Scientific, Waltham, USA) coupled with a Dionex Ultimate 3000 nano-UPLC system. Data was acquired using Xcalibur v4.1.31.9, Chromelean v6.8 (SR13), Orbitrap MS v2.9 (build 2926), and Thermo Foundations 3.1 (SP4). Peptides were dissolved in 0.1% Formic Acid (FA, Sigma-Aldrich, 56302) and 2% acetonitrile (ACN, Burdick & Jackson BJLC015CS) and loaded on a C18 trap column (PepMap100, 9027905000, 300 μm × 5 mm × 5 μm). Approximately 400 ng of peptide was injected. Samples were trapped onto the column and washed for 3 min before the valve was switched and peptides eluted onto the analytical column, as described hereafter. Chromatographic separation was performed with a Waters nanoEase (Zenfit) M/Z Peptide CSH C18 column (186008810, 75 μm × 25 cm × 1.7 μm), as described below. The solvent system employed was solvent A: LC water (Burdick and Jackson BJLC365), 0.1% FA, and solvent B: ACN, 0.1% FA. The multi-step gradient for peptide separation was generated at 300 nL/min as follows: time change, 100 min; gradient change, 5–30% Solvent B; time change, 10 min; gradient change, 30–80% Solvent B. The gradient was then held at 80% Solvent B for 10 min before returning it to 5% Solvent B and equilibrating the column for 15 min. All data acquisition was obtained using Proxeon stainless steel emitters (ThermoFisher Scientific, TFES523). The mass spectrometer was operated in positive ion mode with a capillary temperature of 320 °C. The applied electrospray voltage was 1.95 kV.

### 2.5. Viral Load

Genomic DNA was extracted from SACMV-infected leaf tissue using a CTAB-based method [40] with modifications. CTAB extraction buffer was supplemented with 40 mg/mL polyvinylpolypyrrolidone (PVPP) and 0.2% 2-mercaptoethanol before the buffer was preheated at 65 °C for 10 min. A total of 500 μL of CTAB buffer was added to 0.5 g ground leaf tissue and incubated at 65 °C for 45 min. Chloroform-isoamyl alcohol (24:1 ratio) was added to the samples, followed by gentle mixing and centrifuging at full speed for 10 min. The top aqueous phase was transferred into a new tube and the latter step repeated. Ice cold isopropanol (1:1 ratio) was added and the samples were thoroughly mixed and incubated in the freezer for 45 min. Samples were then centrifuged at full speed for 10 min and the DNA pellet was washed with ice cold 70% and 95% ethanol, respectively, and air-dried. DNA was resuspended in nuclease free water with 1% RNase and incubated at 37 °C for 1 h to allow digestion. Thereafter, DNA was stored at 4 °C overnight, then quantified and stored at −20 °C. DNA (25 ng) was used as a template for SACMV relative viral load quantification using qPCR. Quantification was carried out in triplicates using Maxima SYBR Green/ROX qPCR Master Mix (2X) (ThermoFisher Scientific, Waltham, USA) with SACMV CP primers (Fwd: 5′ ACGTCCGTCGCAAGTAC 3′, Rev: 5′ ATTGTCATGTCGAATAGTACG 3′). SACMV viral loads were calculated using the log 2^(ΔΔCT)^ method [41]. Student’s *t*-test was used to access statistical differences between samples.

### 2.6. Changes in Gene Expression Levels

RNA was extracted from SACMV-infected and Agl1-mock inoculated leaf tissue using QIAzol Lysis Reagent (QIAGEN, Hilden, Germany) according to the manufacturer’s protocol. A RevertAid First Strand cDNA Synthesis Kit (ThermoFisher Scientific) was used to synthesise first strand cDNA using Oligo(dT)18 primer according to the manufacturer’s protocol. Gene expression of selected proteins of interest was measured using relative expression. RT-qPCR was carried out in triplicates with cDNA as a template using Maxima SYBR Green/ROX qPCR Master Mix (2X) with the primers listed in Appendix A. *UBQ10* was used as a reference gene for qPCR quantifications. Gene expression fold change was calculated using the log 2^(ΔΔCT)^ method [41]. Student’s *t*-test was used to access statistical differences genes.

### 2.7. Bioinformatics Analyses

Relative quantification of identified proteins was conducted using Progenesis QI for Proteomics v2.0.5556.29015 (Nonlinear Dynamics, Durham, UK). Data processing included peak picking, run alignment, and normalisation (singly charged spectra were removed from the processing pipeline). Only proteins that contained two or more unique peptides were considered for further analysis. Database interrogation was performed with Byonic Software v3.8.13 (Protein Metrics, Cupertino, USA) using a cassava reference proteome (Cassava_RefProt_UP000091857_25032020.fasta) sourced from UniProt (www.uniprot.org, accessed on 1 August 2021) dated 25/03/2020. Phytozome v13 (https://phytozome-next.jgi.doe.gov/, accessed on 25 July 2021) was used to assess *M. esculenta* v8.1, v7.1, and v6.1 genomes and retrieve protein family data. UniProt and TAIR were used to retrieve theoretical molecular weight and *Arabidopsis* homologs of identified proteins, respectively. STRING version 11.0 (https://string-db.org/, accessed on 26 September 2021) was used to study protein–protein interaction networks. STRING network parameters were selected as follows: (i) Full network; (ii) Confidence edge; (iii) Interaction sources selected were Textmining, Experiments, Databases, and Co-expression; (iv) Confidence levels were between 0.8 and 0.9; (v) More than 50 interactors 1st and 2nd shell; and (vi) Hide disconnected nodes in the network. KEGG parameters were as follows: (i) Search background was selected at the whole genome level; (ii) Statistical test method was Fisher; (iii) Multi-test adjustment method was performed by the method of Benjamini and Hochberg [42]; and Significance level was 0.05. Gene ontology (GO) annotations were carried out on Gene Ontology at TAIR (https://www.arabidopsis.org/tools/bulk/go, accessed on 10 September 2021) using default settings.

### 2.8. CMD Symptom Severity Scoring

The symptom severity score index was used as a guideline for the assessment of CMD symptom development in TME3 and T200 cassava landrace plants following infection with SACMV. The mean severity score was calculated from a total of six leaves from three plants per time point and treatment using a 0–5 score index from Fauquet and Fargette [43] as a guideline for the symptoms assessment, where 0 represent no disease symptoms; 1 represent mild/faint mosaic with mild distortions at bases of most leaves; 2 represent pronounced mosaic pattern on most leaves, leaf malformation, 5% size reduction; 3 represent severe mosaic, distortion, reduced size; 4 represent severe mosaic, severe distortion, up to 50% size reduction; and 5 represent very severe mosaic, distortion, leaf twisting, 50–80% size reduction.

## 3. Results

### 3.1. SACMV Infection Development and Viral Load

CMD characteristic symptoms were observed in both susceptible T200 and tolerant TME3 landraces at 32 and 67 dpi with SACMV (Figure 1a). These include leaf deformation, yellow/pale-green mosaic, and reduced leaf size (red arrows). Agl1 mocked plants did not show any disease symptoms (white arrows). Minor symptoms were observed on newly formed leaves of TME3 at recovery (67 dpi), whereas severe symptoms were still observed in T200 plants. T200 infected plants generally showed severe symptoms, with a significantly high CMD symptoms score of 5 at 67 dpi compared to TME3 infected, which had a lower symptom score of 2.7 (Figure 1b). There was significant difference in the accumulation of SACMV DNA A in T200 and TME3 at 32 and 67 dpi (Figure 1c). T200 had viral loads of 7.7 and 10.4-fold higher relative to *UBQ10* at 32 and 67 dpi, respectively. TME3 had a fold change of 6.8 and 8.0 relative to *UBQ10* at 32 and 67 dpi, respectively, which was lower compared to the fold change in T200.

### 3.2. Protein Identification and Quantification

Leaf tissue proteins were extracted and isolated from T200 and TME3 landraces inoculated with SACMV infectious dimers, DNA-A, and DNA-B. LC-MS analysis and relative quantification resulted in the overall identification of 2682 proteins in both T200 and TME3 at 32 dpi, 49 of which were differentially expressed (DE) with fold change >1.0 (*q* < 0.05) (Table 1, Sheets S1 and S2). In T200, 30 and 19 were under and over-expressed, respectively, while in TME3, 27 and 22 were under and over-expressed, respectively. At 67 dpi, a total of 2817 proteins were identified in both T200 and TME3, with 206 of the proteins differentially expressed with fold change >1.0 (*q* < 0.05) (Table 1, In T200, 63 and 143 were under and over-expressed, respectively, while in TME3, 81 and 125 were under and over-expressed, respectively (Sheets S1 and S2). Ten proteins were expressed in both TME3 and T200 at 32 and 67 dpi. A large number of proteins were not characterised based on the status of the current cassava genome and protein annotations (Manihot esculenta v6.1; phytozome v12.1; https://phytozome.jgi.doe.gov, accessed on 25 July 2021).

Figure 2 shows a heatmap of all DEPs in T200 and TME3 at 32 and 67 dpi. Dendrogram showed differences between leaf proteome at 32 dpi and 67 dpi, with the proteome of T200 at 67 dpi clustering separately from T200 at 32 dpi and TME3 at 32 and 67 dpi. At 67 dpi, proteins that were relatively over-expressed grouped together (red band), while under-expressed proteins grouped together (dark green band). Most over-expressed proteins branched out from the same node at 67 dpi, and the same was observed for under-expressed proteins. Interestingly, at both 32 and 67 dpi, an opposite pattern between susceptible T200 and tolerant TME3 was observed, where over-expression of a group of proteins in one landrace was under-expressed in the other landrace (Figure 2).

### 3.3. Gene Ontology Analysis

Gene ontology analyses were performed on differentially expressed (DE) proteins of T200 and TME3 landraces at 32 and 67 dpi to obtain an overview representation of how SACMV infection affects different processes in the plant. In T200 at 32 dpi, more than 25% of the 19 over-expressed proteins were localised in cellular components such as cytoplasm, mitochondrion, and plastid. A total of 33% of over-expressed proteins involved in stress response were positively induced by SACMV infection, while induced molecular functions included catalytic activity, hydrolase activity, and nucleotide binding, all with more than 30% of over-expressed proteins involved (Appendix A). Of the 30 under-expressed proteins, more than 30% were localized in the chloroplast, cytoplasm, nucleus, and plastid. Stress response-related proteins (30%) and proteins with catalytic activity (32%) were negatively affected by SACMV infection in T200. In TME3, at 32 dpi, of the 22 and 27 proteins that were over-expressed and under-expressed, respectively (Table 1), more than 30% of the 22 over-expressed proteins were found in the cytoplasm and cytosol and were involved in stress response and catalytic and hydrolase activity. The 27 proteins that were under-expressed in TME3 were distributed across several cellular components, including chloroplast, cytoplasm, nucleus, and plastid, and they were involved in biosynthetic processes and catalytic activity (Appendix A).

At 67 dpi, the over-expressed (125) and under-expressed (81) proteins in TME3 in response to SACMV infection (Table 1) were distributed across different GO groups (Appendix A); however, only over-represented GO annotations with at least 30% of DEPs are presented in Figure 3. SACMV infection induced an increase in the expression of proteins with functions in catalytic activity and binding, which are involved in biological processes such as stress response and biosynthetic processes. These over-expressed proteins were localised in the plastid, cytosol, cytoplasm, and chloroplast (Figure 3a). Under-expressed proteins in TME3 at 67 dpi were localised in the nucleus, cytosol, cytoplasm, and chloroplast. These were involved in metabolic and cellular processes and had binding functions (Figure 3a). In T200, 143 proteins were over-expressed and 63 under-expressed at 67 dpi (Table 1). These proteins are also annotated across a range of GO groups (Appendix A). Over-expressed proteins in T200 had catalytic and binding functions and were involved in biosynthetic, cellular, and metabolic processes and response to stress and chemical response. These proteins were found in the cytosol, cytoplasm, and chloroplast (Figure 3b). More than 30% of proteins were repressed by SACMV in T200, and these proteins had catalytic functions; were involved in metabolic and cellular processes; and localised in the nucleus, extracellular region, cytosol, cytoplasm, and chloroplast (Figure 3b).

### 3.4. KEGG, Protein–Protein Interaction Network and Reactome Pathway Analysis

Proteins do not function alone, and therefore the metabolic pathways and network protein partners associated with the differentially expressed proteins in response to SACMV were explored. KEGG analysis of DEPs in T200 and TME3 at 32 dpi demonstrated 6 significantly enriched pathways, whereas 22 pathways were significantly enriched at 67 dpi in response to SACMV infection compared to mock-inoculated (Table 2, Appendix A). KEGG pathways that were unique to 67 dpi included nitrogen metabolism, pyruvate metabolism, glycolysis/gluconeogenesis, ribosome, and proteasome. STRING applies known and predicted protein–protein interactions extracted from genomic context predictions, high-throughput lab experiments, co-expression, automated text mining, and previous knowledge in databases to create interaction networks. At 32 dpi, the differentially expressed protein–protein network (*p*-value < 1.0 × 10^−16^) revealed four major functional groups: DNA repair and replication, secondary metabolites biosynthesis, glycolysis, and plant innate immunity (Figure 4a). Proteins such as suppressors of the G2 allele of *skp1* (SGT1) subunit A and SGT1B were some of the interactors with functions in plant innate immunity. SGT1 directly interacts with heat shock protein 90 (HSP90) and *Arabidopsis* homolog of Manes.09G037400 (AT3G12050; activator of Hsp90 ATPase family), which was under-expressed in both TME3 and T200 with ratios of 1.61 and 1.48, respectively (Figure 4a; Table 3). SGT1 is also an essential S-phase kinase-associated protein (SKP1)-interacting eukaryotic protein which is part of the Cullin1-based RING ligases SCF complexes that function in plant-virus interactions. At 67 dpi, proteins mainly formed two groups: proteasome and translation and translation regulation (Figure 4b). Regulatory particle triple-ATPase (RPT) subunit 2a and RPT2b directly interacted with Manes.01G032500 (AT1G29150, Proteasome component family) and Manes.05G155000 (AT1G45000, Holliday junction DNA helicase RuvB P-loop family), which were both over-expressed in T200 and under-expressed in TME3 (Figure 4b, Table 3). Protein–protein interaction network functional enrichment analysis of over-expressed proteins in TME3 at 67 dpi revealed an interaction with Sac52 (Appendix A), an important plant defence response protein. At 32 dpi, metabolism (six proteins) was the only pathway that was significantly enriched (FDR < 0.05). Three pathways were significantly enriched at 67 dpi: metabolism (28 proteins), L13a-mediated translational silencing of Ceruloplasmin expression (eight proteins), and antigen processing-Cross presentation (three proteins).

### 3.5. Differentially Expressed Protein Groups in Response to SACMV Infection at 67 dpi

Because the molecular mechanisms of tolerance to CMD in TME3 are of key interest, we chose to focus on and compare responses to SACMV in T200 and TME3 at 67 dpi, which is when recovery to infection in TME3 has already begun. Differentially expressed proteins between mock-inoculated and SACMV-infected leaves in T200 and TME3 at 67 dpi are presented in Table 3.

#### 3.5.1. Metabolic Pathways

A total of 56 proteins were involved in primary metabolic pathways in TME3 infected with SACMV at 67 dpi (Table 2). Functions of up- and under-expressed DEPs in this group included amino acid metabolism, secondary metabolism, photorespiration, and carbohydrate metabolism. Primary metabolites are crucial for the normal development of the plant. In T200, glycine cleavage system H proteins (Manes.07G087900, Manes.15G169800) and aminotran_5 domain-containing protein (Manes.01G181700), also functioning in the biosynthesis of secondary metabolites, were under-expressed. In addition, phosphoglycerate kinase (PGK, Manes.14G009000), a major role player in energy generation, was also under-expressed (Table 3). In *Nicotiana benthamiana*, chloroplast PGK was found to be one of the proteins that binds to bamboo mosaic virus (BaMV) RNA [45]. Trehalose-6-phosphate synthase 7 (Manes.16G042700) protein plays a role in plant growth, development, and regulation of defence response against pathogens, and this protein was over-expressed in T200 and under-expressed in TME3. Other proteins involved in the biosynthesis of metabolites include Inositol-1-monophosphatase, Phosphoacetylglucosamine mutase, AA-kinase domain-containing protein, and Arginine biosynthesis bifunctional protein, to mention a few (Table 3). One uncharacterised protein, Manes.09G142200, belonging to the family Enoyl-(Acyl carrier protein) reductase, was highly over-expressed by 67-fold in TME3 and under-expressed (1.63-fold) in T200.

#### 3.5.2. Biosynthesis of Secondary Metabolites

A total of 36 proteins were involved in the biosynthesis of secondary metabolites (Table 2). Plant hosts require secondary metabolites for several roles, including pigmentation, growth, reproduction, and resistance to pathogens, amongst many others. Peroxidase protein, Manes.15G103900, was over-expressed in T200 and under-expressed in TME3 (Table 3). Peroxidases are known to play important roles in oxidative stress, auxin catabolism, and cross-linking of structural cell wall proteins, amongst other things. In this study, a Methyltransferase protein (Manes.05G142300) was under-expressed almost seven-fold in TME3 and over-expressed three-fold in T200 (Table 3). 3-ketoacyl-CoA synthase (Manes.16G087600) and AB hydrolase-1 domain containing proteins (Manes.12G095700, Manes.04G110900, Manes.05G029200, and Manes.13G092100) are some of the structural proteins that play a role in lipid homeostasis (Table 3) [46,47]. Other characterised proteins in this group include PALP domain-containing protein, Phospholipase D, Serine hydroxymethyltransferase, FAD-binding PCMH-type domain-containing protein, GPAT_N domain-containing protein, SAICAR_synt domain-containing protein, Inositol-1-monophosphatase, and F420_oxidored domain-containing protein (Table 3). Some of the proteins, however, were not characterised, but these belonged to protein families including KR domain (Manes.10G026600), p450 (Manes.06G006200 and Manes.09G183600), and aminotransferase_3 (Manes.13G089400), to mention a few.

#### 3.5.3. Glycolysis/Gluconeogenesis

This pathway is associated with the generation of energy. When a plant is attacked by an external stimulus, it direct most of its energy to defence response mechanisms and less to cellular metabolic pathways. In this functional group, five proteins were involved. Three proteins were over-expressed in both T200 and TME3, and these are also involved in metabolic pathways: FBPase domain-containing protein (Manes.08G083500), pyrophosphate-fructose 6-phosphate 1-phosphotransferase subunit alpha (Manes.18G001100), and Manes.04G146300 (uncharacterised). FBPase (fructose 1,6-bisphosphatase) catalyses the hydrolysis of fructose 1,6-phosphate to fructose 6-phosphate in gluconeogenesis and the Calvin cycle. The reverse is true for glycolysis with phosphofructokinase (PFK) and pyrophosphate-fructose 6-phosphate 1-phosphotransferase subunit alpha enzymes catalysing the reaction. With FBPase over-expressed, it means that the plant had enough cellular ATP concentration and there was no need to produce more in both T200 and TME3 at 67 dpi. Uncharacterised Manes.04G146300 protein originates from the family glycoside hydrolase family 19, which consists of chitinase class I. Phosphoglycerate kinase (Manes.14G009000) was over-expressed in TME3, with a ratio of 1.51, and under-expressed in T200, with a ratio of 1.12 (Table 3). This enzyme catalyses the high-energy phosphoryl transfer of the acyl phosphate of 1,3-bisphosphoglycerate to ADP to produce ATP [48]. The last uncharacterised protein in this group, Manes.08G094400, belongs to the thiamine pyrophosphate enzyme (TTP) family, and it was only over-expressed in T200 (ratio 1.91; Table 3).

#### 3.5.4. Ribosome

Ribosomal proteins are known to be components of the protein synthesis machinery. However, with recent studies, it would seem that these proteins have extra functions such as stress signalling [49]. Seven DEPs were grouped to the ribosome pathway (Table 3), and three of these are uncharacterised. Three proteins were under-expressed in T200 and over-expressed in TME3, three over-expressed in T200 and under-expressed in TME3, and one was under-expressed in both landraces. For example, ribosomal_L14e domain-containing protein (Manes.03G162900) was over-expressed with ratio of 4.36 in T200 and under-expressed with ratio of −1.86 in TME3 (Table 3). This protein binds to the 60S ribosomal subunit and plays a role in translation [50]. The 40S ribosomal protein S4 (Manes.03G095200) had ratios of 1.59 in T200 and −1.06 in TME3. An increase in transcript levels of this protein was observed in *Vanilla planifolia* Jacks infected with fungi [51], whereas there was a decrease in transcript in *Arabidopsis* 48 h after inoculation with *A. tumefaciens* [52]. The 60S ribosomal protein L13 (Manes.07G017100) was under-expressed in T200 (−3.86) and over-expressed in TME3 (2.11). This protein has been shown to interact with CMV P6 (translation re-initiator) in *A. thaliana* [53]. Uncharacterised proteins, Manes.10G151400 (−3.90 ratio in T200 and −1.14 in TME3), Manes.17G064800 (−4.61 in T200 and 1.83 in TME3), and Manes.10G019200 (3.30 in T200 and −1.16 in TME3) are members of the ribosomal protein family L3, L32, and L35Ae. The last protein of interest in this group was ribosomal_L7Ae domain-containing protein (Manes.12G139100), which was under-expressed in T200, with a ratio of 5.24, and over-expressed in TME3, with the ratio of 2.18 (Table 3). Much like the other proteins in this group, very little has been reported on this protein; however, its molecular function includes RNA binding.

#### 3.5.5. Proteasome

Proteasomes form a large portion of the ubiquitin proteasome system, responsible for degrading the intracellular proteins involved in most cellular metabolic processes. Three proteins, also involved in metabolic pathways, with functions in proteasomes, were only over-expressed in T200. PCI domain-containing protein (Manes.01G032500) was over-expressed by up to seven-fold compared to mock sample in T200, and under-expressed in TME3 at 67 dpi. PCI (Proteasome, COP9 signalosome, and initiation factor 3) domain mediates and stabilizes protein–protein interactions within multi subunit protein complexes [54]. The other two proteins were both AAA domain-containing proteins with ratios of 4.75 in T200 and −1.32 in TME3 (Manes.13G043200) and 2.82 in T200 and −1.07 in TME3 (Manes.05G15500) (Table 3). ATPase associated with various cellular activities (AAA) plays a role in the cell, including cell-cycle regulation, protein proteolysis, and disaggregation, organelle biogenesis, and intracellular transport. 

#### 3.5.6. Transcription of Selected Differentially Expressed Proteins

Based on proteomics analysis, ten proteins that were differentially expressed in response to SACMV were selected for transcription analyses using RT-qPCR (Figure 5). These selected proteins have functions in disease response and cell morphogenesis and were differentially expressed in both landraces, two at 32 dpi (C3HC4-type RING finger and LRR) and eight at 67 dpi (kinase, *SnRK1*.*1*, two receptor-like kinases, *LRX2*, receptor-like protein, two pathogenesis-related proteins). At 32 dpi, RT-qPCR results demonstrated that C3HC4-type RING finger (Manes.08G075100) was expressed with fold change of 0.5, whereas LRR had a fold change of −1.7 compared to the mock-inoculated control (Figure 5). Transcription levels of the latter genes did not reflect protein translation expression regulation (Table 3). At 67 dpi, all eight genes were demonstrated to be over-expressed with fold changes ranging from 1.7 to 10.8 higher than the mock-inoculated control, with pathogenesis-related protein_2 (Manes.15G007900) showing the highest expression levels (Figure 5). Transcripts of receptor-like kinases (Manes.05G194900, Manes.14G124000), Kinase (Manes.08G126700), and receptor-like protein (Manes.06G055700) were over-expressed with fold change of 7.9, 3.8, 1.7, and 2.5, respectively (Figure 5), whereas the expression levels of their proteins were under-expressed based on the proteomics results (Table 3). The results herein suggest differences between transcript and protein levels may be due to post-transcriptional regulatory processes.

## 4. Discussion

Cassava is an economically important crop and a major source of nutrition in the tropical and sub-tropical regions of Africa and Asia, and it additionally shows tolerance to drought. This makes it a targeted crop in future climate change where temperatures are expected to rise. However, cassava production is threatened by viral diseases such as CMD, which is caused by a number of viruses, including geminiviruses. As such, it is important to identify proteins that are differentially expressed in response to SACMV as these will help bridge the gap that exist in understanding plant-virus interactions in this and other orphan crops, and furthermore inform engineering strategies for crop varieties that will better tolerate and/or resist virus diseases. This study reports the first SACMV-cassava proteome map comparing susceptible and tolerant landraces and further contributes new information to the proteomic data pool. Finding more clues to the molecular mechanisms of tolerance to SACMV in TME3 at 67 dpi is also of paramount interest. Several important GO processes in response to SACMV in TME3 were identified herein that may play a role in CMD tolerance. In particular, we also focussed on individual proteins shown to be involved in geminivirus responses in other plants that may also play a role in cassava. While it is well documented that protein modifications are a consequence of pathogen infection of plants, fewer studies have been performed on virus infections, in particular on geminiviruses. It is noteworthy that 8 out of the 18 genes involved in processes related to protein modification or protein translation and metabolism (such as ubiquitination, rubylation, phosphorylation, acetylation, or protein folding) were differentially expressed in *N. benthamiana* infected with the geminivirus tomato yellow leaf curl Sardinia virus (TYLCSV) [55]. It is therefore not surprising that, in SACMV-infected T200 and TME3, seven of the DEPs were related to the ribosome pathway. Ribosomal proteins are not only known to be components of the protein synthesis machinery, but are also involved in stress signalling [49]. In cassava infected with Indian cassava mosaic virus (ICMV) and Sri Lankan cassava mosaic virus (SLCMV), components of the 80S ribosome and 40S small subunit were upregulated in susceptible cassava cultivar H226 [32]. In a proteome study of the begomovirus, tomato chlorotic mottle virus (ToCMoV), in resistant tomato, two ribosomal proteins (40S ribosomal protein S7 and ribosomal protein L11/12) were also differentially expressed [25]. These results suggest that translational control via ribosome activity may be a general response to geminivirus-induced stress. Because T200 and TME3 showed contrary expression (up or down) in six ribosomal proteins, we conclude that these play a role in susceptibility and tolerance in T200 and TME3, respectively. Perturbations in protein translation also reflect dual expression of virus and plant host proteins induced during their interaction.

Notably, a large number of the SACMV-responsive cassava DEPs annotated by GO were located in different cellular components, but largely in the cytoplasm and plastid, including the chloroplast and thylakoid (Figure 3, Appendix A). From this observation, we conclude that chloroplasts may be involved in response to SACMV. Plastids have versatile roles providing essential metabolic and signalling functions, and affect photosynthesis, plant growth, and development. They also play role in plant-microbe interactions [56]. Chloroplasts have more recently been shown to have other functions, including involvement in plant defence responses through the production of reactive oxygen species, defence-related hormones, and plastid-to-nucleus retrograde signalling pathways [57]. Together with the fact that chloroplasts are central hubs for facilitating communication with different plant cellular compartments, chloroplasts functions have become targets of plant pathogens. Pathogens deliver effector proteins that disrupt the normal functioning of the chloroplast and result in suppressed defence responses and subsequent infection [58]. We propose that the identified differentially expressed chloroplast proteins in our study play a role in virus tolerance and susceptibility in cassava. The majority of over-expressed proteins (56%) in TME3 at 67 dpi were localized in the chloroplast (Figure 3a), whereas 31% of these chloroplast proteins were under-expressed in T200 at 67 dpi (Figure 3b). Manes.15G029400 *Arabidopsis* homolog AT4G14880, a chloroplast protein whose gene expression is mediated by abscisic acid (ABA), was over-expressed in TME3 (1.25 ratio) at a symptom recovery stage (67 dpi). ABA is an essential signal for plant resistance to pathogens, and it is synthesized in the chloroplast. The over-expression of Manes.15G029400 indicates that ABA production in the chloroplast was not affected by SACMV infection. We conclude that ABA-mediated defence mechanisms are active in TME3 and play a role in the reduction of SACMV replication and symptom reduction at 67 dpi. Results from other studies have shown the importance of ABA in biotic stress and host defence against viral pathogens. For example, in *N. benthamiana*, it was proved that *Chinese wheat mosaic virus* suppressed the ABA pathway, which was important in inducing plant host defence against this virus [59]. Although the contribution of these chloroplast-associated proteins to tolerance is not currently known, we suggest that over-expressed chloroplast proteins in tolerant TME3 at 67 dpi play a role in virus and symptom recovery phenotype (Figure 1).

Coat protein (CP) from a number of viruses, including TMV, *Potato virus X*, and CMV, has been shown to accumulate in the chloroplast of their host and cause chloroplast ultrastructure disruption as well as the development and severity of mosaic or chlorosis symptoms caused by viral infections [60,61,62]. TME3 landrace shows a recovery phenotype and significant reduction of SACMV DNA accumulation at 67 dpi (Figure 1), compared with susceptible T200 landrace. A decrease in DNA also implies that virus coat protein synthesis is decreased as these are co-regulated, which suggests that a decrease in CP in chloroplasts in TME3 at 67 dpi could contribute to symptom attenuation at the recovery stage. It was also reported that some of these viral CPs are associated with the thylakoid [63,64]. In this study, a relatively high number of over-expressed proteins (15; 13%) were localised in the thylakoid at 67 dpi in TME3 (Figure 3a). Proteins such as Manes.10G027600 (Rubisco activase; 6.89 ratio) and Manes.14G009000 (Phosphoglycerate kinase 1; 1.15 ratio) were amongst the over-expressed proteins localised in the thylakoid (Table 2) and are known to interact with virus nucleic acids or proteins [58]. A difference that was seen between T200 and TME3 at 67 dpi was the presence of the Manes.08G136300 (linker histone; 1.08 ratio) protein involved in epigenetic regulation of gene expression in TME3 (Table 1, Figure 3a). Linker histones are major negative regulators that limit the accessibility of DNA to various trans-acting factors, which in turn enables epigenetic suppression of genes [65]. It is not clear how the mechanism of this protein would contribute to SACMV tolerance phenotype in TME3; however, further studies on this protein are warranted.

Additionally, of particular interest with regard to SACMV DNA accumulation was the upregulation and downregulation of the SUMO-activating enzyme subunit in T200 and TME3 at 67 dpi, respectively (Table 1). SUMOylation is an essential post-translational modification in plants and is also affected by oxidative stresses. Geminiviruses replicate in the nuclei of infected plant cells using the plant DNA replication machinery, including proliferating cellular nuclear antigen (PCNA), a cofactor that orchestrates genome duplication and maintenance by recruiting crucial players to replication forks. Posttranslational modification of PCNA by SUMO plays an essential role in the switching of PCNA between interacting partners during host replication, recombination, and repair mechanisms. The geminivirus Rep induces the accumulation of the host replication machinery by interfering with the cell cycle and impairs SUMO conjugation of proliferating cellular nuclear antigen (PCNA) and the SUMO E2 conjugation enzyme (SCE1) [66]. The significant down-regulation (−5.5 ratio) of the SUMO-activating enzyme subunit in T200 would lead to impairment of SUMO conjugation, thereby switching on the host cell machinery for its own replication. This would contribute to the persistent high SACMV replication levels in T200 at 67 dpi (Figure 1). In contrast, over-expression (4.65 ratio) of SUMO-activating enzyme in TME3 would contribute to a reduction in SACMV replication, leading to recovery at 67 dpi.

At 32 dpi (systemic infection stage), virus replication and symptoms are high in both landraces. At this stage, STG1A and the molecular chaperone HSP90.1 were over-expressed. These proteins directly interact with AT3G12050, a homolog of Manes.09G037400 from the Activator of the Hsp90 ATPase family (Figure 4a). SGT1 is an essential S-phase kinase-associated protein (SKP1)-interacting eukaryotic protein which is part of the Cullin1-based RING ligases SCF complexes that function in plant-virus interactions. SGT1 also binds to HSP90, resulting in the positive regulation of disease resistance conferred by many resistance (NB-LRR) proteins in plants [67]. In *N. benthamiana*, a study found that SGT1a controls the abundance of an R protein Rx after the silencing of SGT1a resulted in the reduction in steady-state levels of Rx. It is understood that any compromise in the activity of HSP90 and levels of SGT1a leads to reduced accumulation of R proteins [67,68]. This is in agreement with what was observed in this study, as Hsp90 activator was under-expressed in both T200 (−1.48 ratio) and TME3 (−1.61 ratio), which may have compromised HSP90. The implications of this would be the observed severe CMD symptoms (Figure 1a,b) as well the accumulation of SACMV DNA (Figure 1c) at 32 dpi. Moshe et al. [69] also proved the roles that SGT1a and HSP90 play in susceptibility to TYLCV in infected tomatoes. Interestingly, in ICMV and SLCMV infected susceptible cassava, a member of the HSP70 chaperones, namely chaperone DNAj, was upregulated. It has been suggested that chaperones may be involved in cell-to-cell movement via modified plasmadesmata (PD), and molecular chaperones are known to be involved in modification of the size exclusion limit of PD [28]. This makes a case for HSP90.1 involvement in virus movement in T200 and TME3 at 32 dpi, corresponding to the systemic infection and high virus loads observed in both landraces.

At 67 dpi in TME3, two overexpressed unique protein interactors, RPT2 and Sac52 (Figure 4b), that we propose are involved in SACMV recovery, are discussed. Firstly, RPT2 is a member of the proteasome regulatory subunits, known to be of importance in defence response [70]. *Arabidopsis* homologs of Manes.01G032500 (AT1G29150; 7.14 ratio in T200 and −1.40 ratio in TME3) and Manes.05G155000 (AT1G45000; 2.82 ratio in T200 and −1.07 ratio in TME3) (Table 3) interact directly with RPT2a and RPT2b proteins (Figure 4b). RPT2a has been linked with PAMP triggered immunity (PTI) and the establishment of broad-spectrum pathogen resistance and systemic acquired response in *Arabidopsis* in response to bacterial infections. With the up-regulation of proteins directly interacting with RPT2a, one would propose that there would be the reduction in systemic infection due to RPT2a-induced responses. However, this is not the case for susceptible landrace T200, which shows no recovery phenotype at 67 dpi as disease symptoms are still severe and SACMV accumulation is significantly high (Figure 1). Interestingly, a study showed that *Cauliflower mosaic virus* relies on the proteasome subunits RPT2a and RPT2b for robust infection [71], indicating that these proteasome subunits could have different functions in different host-virus interactions. This observation supports the results observed in our study, where higher expression of the proteasome interactors was linked to T200 with a greater virus load and down-regulation linked to TME3 (Appendix A) with low virus accumulation and a recovery phenotype (Figure 1, Table 3).

The second significant protein of interest is Sac52, also known as the ribosomal protein L10 (RPL10), which interacts with Ribosomal_L7Ae domain-containing protein homolog (Manes.12G139100, AT1G77940), which was over-expressed uniquely in TME3 only at symptom recovery (67 dpi) (Table 3, Appendix A, Sheet S2). RPL10 is a known key player in the NIK1-mediated antiviral immunity defence response to geminivirus infection as it plays the role of viral effector detection, triggering effector triggered immunity (ETI) mechanisms in plants [72]. This defence mechanism is established with interaction between membrane-bound LRR receptor-like serine/threonine kinase and its virulence target nuclear shuttle protein (NSP), found on DNA B component of the bipartite geminiviruses. During geminivirus infection, NIK1 autophosphorylates the kinase domain on threonine-469 and threonine-474, resulting in NIK1 kinase activation [9,73. Once activated, NIK1 indirectly phosphorylates the cytoplasmic RPL10, which is then subsequently translocated to the nucleus where it interacts with the L10-interacting MYB domain-containing protein (LIMYB). This interaction mounts a defence response in the form of translational inhibition of several host genes and decreased geminiviral transcript association with polysomes, consequently negatively impacting virus infection. With regard to the role that RPL10 may play in TME3 tolerance to SACMV, we propose that this protein may be expressed in abundance and then phosphorylated by membrane-bound NIK1, which would then lead to downstream translational suppression of viral proteins and thus virus movement, which would contribute to the symptom and virus recovery phenotype observed in TME3 (Figure 1a). This is in agreement with observations in other different plant hosts [73,74,75,76,77,78]. It is also important to note that the RPL10 interactor (Manes.12G139100) is located on chromosome 12, known to be the location of the QTL for dominant CMD2 resistance [79,80,81], thus supporting the likelihood of this protein being a key player in TME3 tolerance to SACMV.

Ubiquitination of geminiviral proteins can contribute to plant defence, and geminiviruses often subvert or hijack ubiquitin-proteasomal pathways [9]. At 67 dpi, one of the highly represented functional group proteins was the proteasome (Figure 4b). The PCI domain-containing protein (Manes.01G032500), involved in the COP9 signalosome, was over-expressed (7.14-fold ratio) compared to the mock in T200, suggesting a major role in susceptibility via interference of this complex. Plant geminiviruses redirect ubiquitination by interfering with the activity of the CSN (COP9 signalosome) complex. The geminivirus C2 protein has been demonstrated to compromise CSN activity on CUL1. Among several responses regulated by the CUL1-based SCF ubiquitin E3 ligases, is response to jasmonic acid (JA), ethylene (ET), and abscisic acid (ABA), known to function in defence responses. A C3HC4-type RING finger protein, Manes.08G075100, was under-expressed in TME3 at 67 dpi and over-expressed in TME3 at 32 dpi (Table 3), supporting a strong role for ubiquitination in TME3 tolerance. This RING finger protein is an E3 ubiquitin ligase (E3L) that is a component of the ubiquitin proteasome system (UPS), with many of their functions involved in plant stress resistance [82,83,84,85]. Recently, it was shown that CRISPR knockdown of a E3L in cassava TME3 protoplasts resulted in an increase in SACMV replication, suggesting that E3L target proteins may play a role in SACMV tolerance [82]. Ubiquitin activating enzyme *UBA1* and *RING-type E3 ubiquitin ligase* are also reported by [55] to be trans-activated by the geminivirus C2 protein. Further evidence for the role of the ubiquitin-proteasome in response to viral infection was recorded in a proteome study on ToCMoV resistant tomato, which showed decreased levels of ubiquitination pathway-related proteins [25].

## 5. Conclusions

SACMV is a highly pathogenic virus in susceptible hosts, leading to major yield losses. In perennial hosts such as cassava, tolerance may represent accommodation of persistent virus infection throughout its life span [86]. Our proteome results identified the proteins involved in different metabolic pathways, some with functions in defence responses to SACMV. We identified proteins in the ribosome, chloroplast, ubiquitination, and cytoplasm GOs, and we propose that these proteins and their connected networks are contributors to SACMV and symptom recovery in TME3. One of these identified proteins of importance is Sac52 (RLP10), which interacts with NIK1 and may contribute to an anti-virus response in cassava. This is highly significant because this finding provides further evidence that PAMP triggered immunity (PTI) may be involved in resistance to plant geminiviruses. While ETI has long been recognized as an efficient specific defence against viruses, PTI characterized in non-viral pathogen-plant interactions has not been found to operate against plant viruses until recently [87]. Geminivirus nucleic acids were recently shown to act as viral pattern associated molecular patterns (PAMPs) and activate transmembrane receptor NIK1, which shares PTI co-receptors regulatory mechanisms for activation [87]. However, geminiviral PAMPs and their cognate PRRs have yet to be identified, and investigation of antiviral PTI mechanisms in cassava is highly warranted. In the cassava TME3 landrace, we conclude that tolerance allows the virus to accumulate to some degree without causing significant loss of fitness, implying that the virus and host co-exist. The results herein support the proposal that symptom recovery in TME3 may be associated with stress tolerance and maintenance of cellular homeostasis [85]. In perennial crops, this would be advantageous as long-term defence responses are energy-intensive. SACMV-tolerance in TME3 would mitigate energy costs, leading to growth recovery and a reduction in symptoms and virus load. The results from this study form the foundation for further *in planta* functional studies to understand cassava tolerance to geminiviruses.

## Figures and Tables

**Figure 1 proteomes-09-00041-f001:**
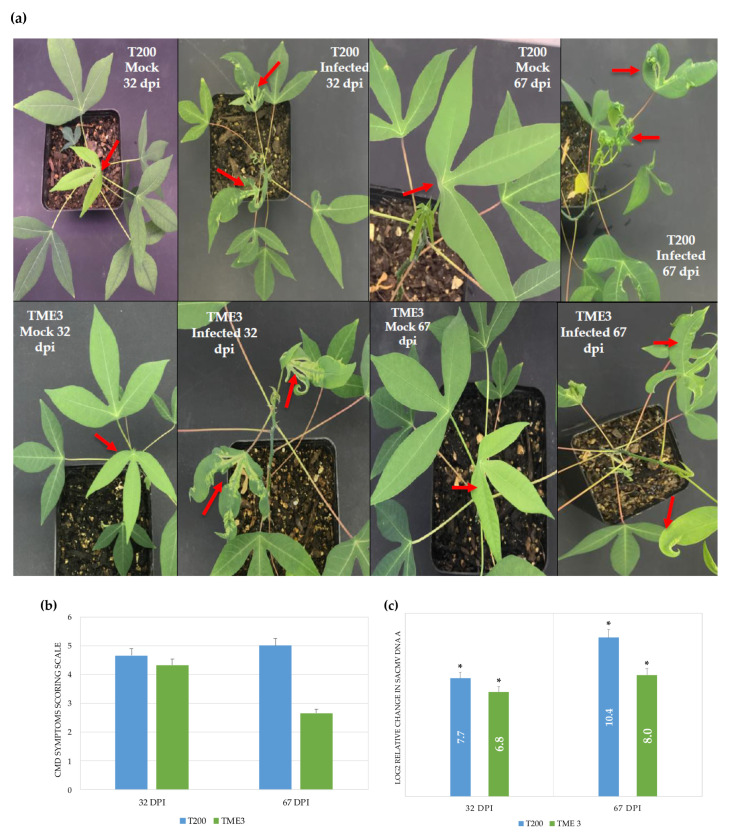
SACMV symptoms scores and viral load quantification in T200 and TME3 landraces at 32 and 67 dpi. (**a**) Characteristic CMD symptoms on T200 and TME3 compared with mock-inoculated plants. The white arrow points at CMD-free leaves, while the red arrow points at CMD-infected leaves. (**b**) CMD symptoms scoring on infected plants. (**c**) Relative change (log 2^(ΔΔCT)^) of SACMV DNA A to internal reference gene *UBQ10* in infected plants. Error bars represent means ± S.D. (n = 3). Asterisks indicate statistically significance at *p* ≤ 0.05 using *t*-test. Symptoms Scoring Scale: 0 = No disease symptoms; 1 = Mild/faint mosaic with mild distortions at bases of most leaves; 2 = Pronounced mosaic pattern on most leaves, leaf malformation, 5% size reduction; 3 = Severe mosaic, distortion reduced size; 4 = Severe mosaic, severe distortion, up to 50% size reduction; 5 = Very severe mosaic, distortion, leaf twisting, 50–80% size reduction.

**Figure 2 proteomes-09-00041-f002:**
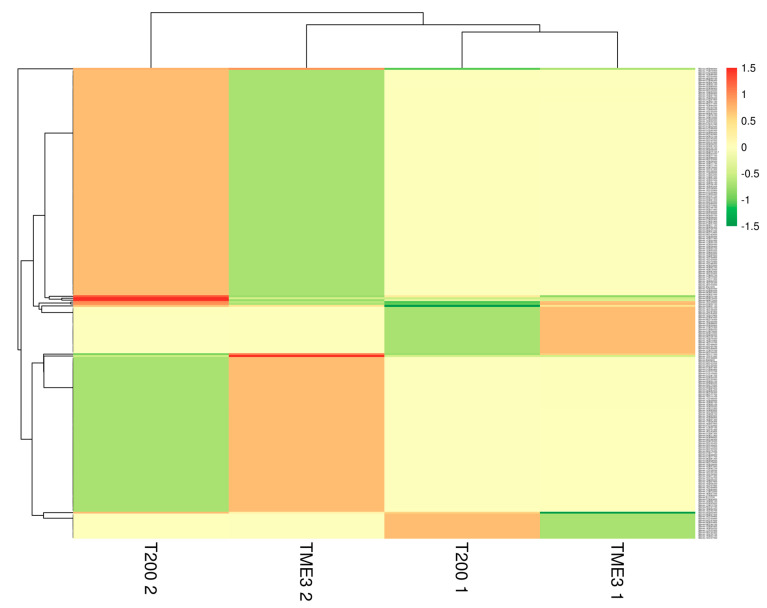
Heatmap of TME3 and T200 differentially expressed proteins at 32 and 67 dpi. 1 represents 32 dpi and 2 represents 67 dpi. ClustVis [44] was used to generate the heatmap. Rows are centred; unit variance scaling is applied to rows. Imputation is used for missing value estimation. Both rows and columns are clustered using Euclidean distance and average linkage.

**Figure 3 proteomes-09-00041-f003:**
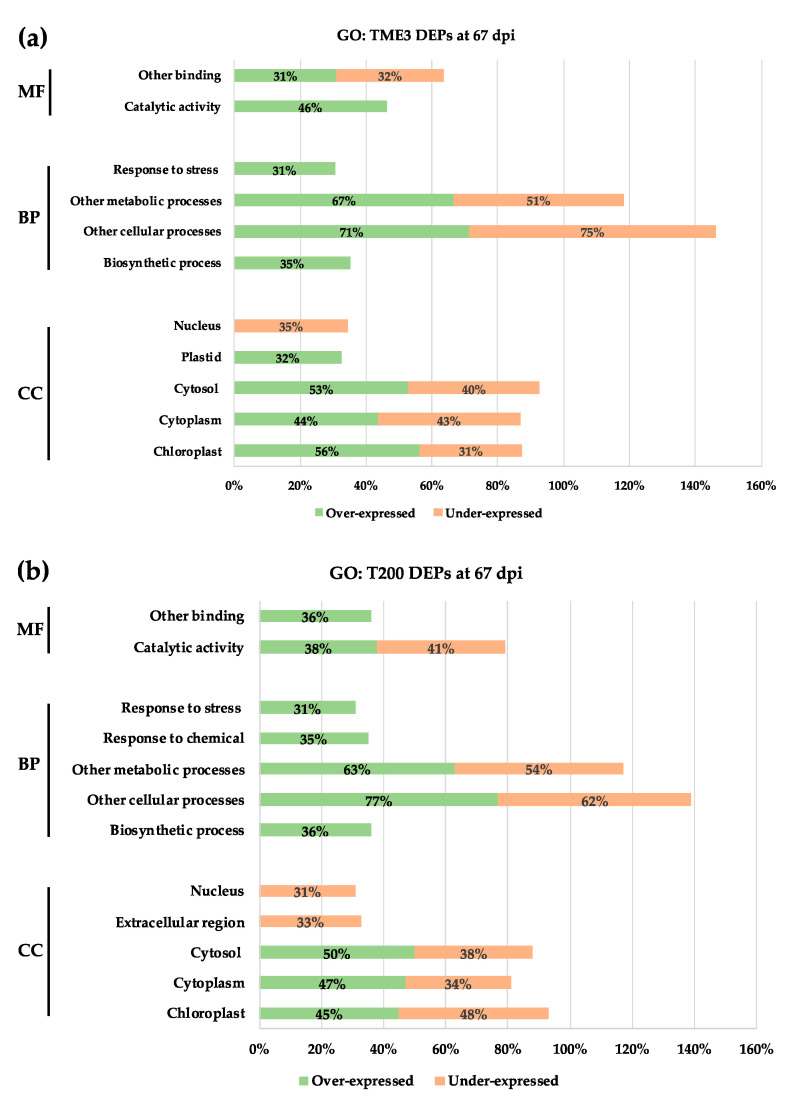
Gene ontology analysis of over-represented differentially expressed proteins from TME3 and T200 at 67 dpi. (**a**) Functional grouping of TME3 proteins. (**b**) Functional grouping of T200 proteins. MF represents molecular function; BP, biological process; CC, cellular component.

**Figure 4 proteomes-09-00041-f004:**
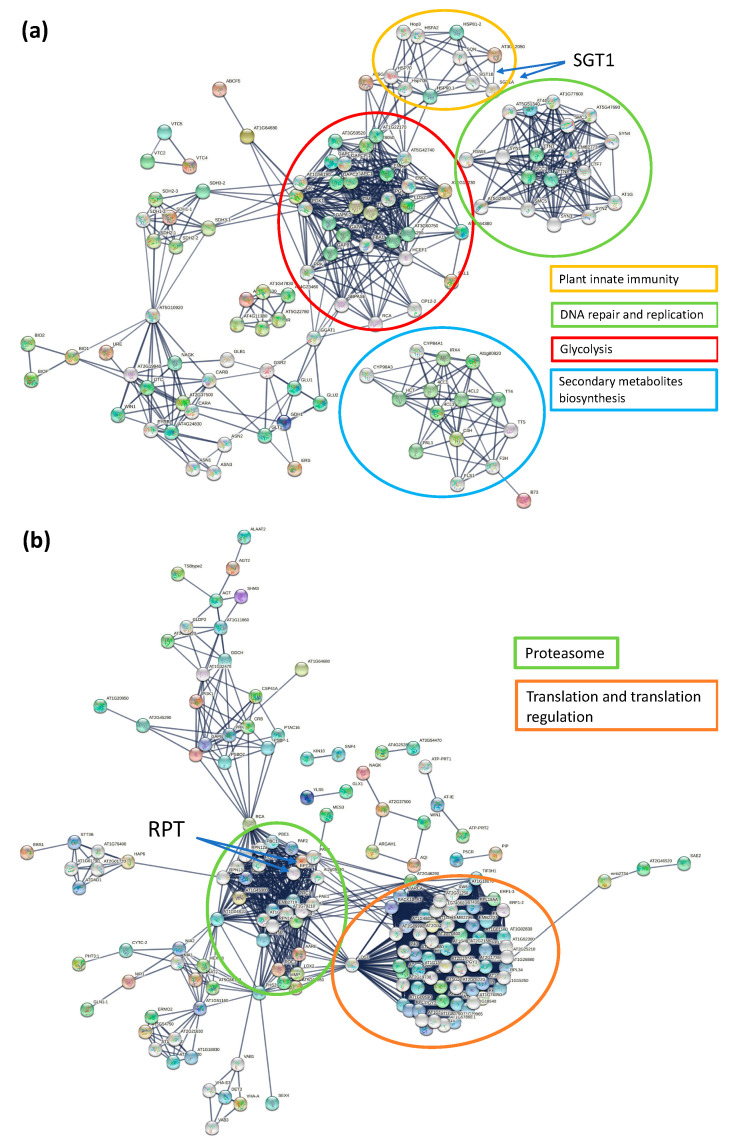
STRING protein–protein interaction network of differentially expressed proteins with *p*-value < 1.0 × 10^−16^ in T200 and TME3 at 32 dpi (**a**) and 67 dpi (**b**). Network Settings: full network; confidence edge; textmining, experiments, databases, co-expression interaction sources; confidence of 0.8 and 0.9, respectively; more than 50 interactors 1st and 2nd shell; hide disconnected nodes in the network.

**Figure 5 proteomes-09-00041-f005:**
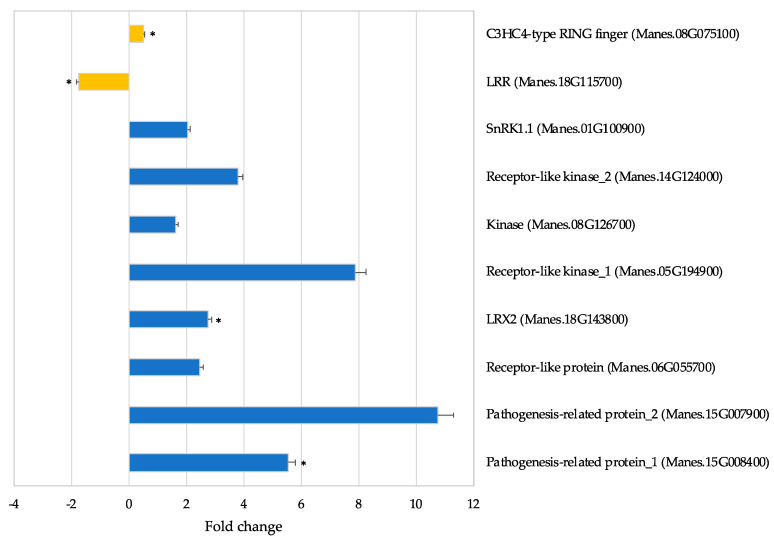
Validation of cassava gene expression levels by real-time RT-qPCR in TME3. Gold represents 32 dpi; Blue represents 67 dpi. Readings are averaged from three biological replicates normalized against endogenous gene *UBQ10*. Error bars represent means ± S.D. (*n* = 3). Asterisks indicate statistically significance at *p* ≤ 0.05 using *t*-test.

**Table 1 proteomes-09-00041-t001:** Summary of differentially expressed proteins in SACMV-infected T200 and TME3 plants compared with mock-inoculated plants at 32 and 67 dpi.

Timepoint	Regulation	Protein Count	Protein Characterisation	GO Categories ^a^	Functional Groups
32 dpi	Under-expressed	30 T200	14 characterised 35 uncharacterised	49 CC	19
		27 TME3		42 BP	35
	Over-expressed	19 T200		36 MF	13
		22 TME3		49 KEGG	11
		**Total** **=** **49**			
67 dpi	Under-expressed	63 T200	128 characterised 78 uncharacterised	198 CC	21
		81 TME3		178 BP	44
	Over-expressed	143 T200		198 MF	23
		125 TME3		199 KEGG	22
		**Total** **=** **206**			

^a^ Categories based on cellular component, CC; biological process, BP; molecular function, MF; and KEGG analysis of the total number of differentially expressed proteins.

**Table 2 proteomes-09-00041-t002:** KEGG analysis of differentially expressed proteins in SACMV-infected T200 and TME3 plants at 32 and 67 dpi.

Timepoint	Pathway	Protein Count	FDR ^a^
32 dpi	Metabolic pathways	13	2.30 × 10^−4^
	Biosynthesis of secondary metabolites	8	2.30 × 10^−3^
	Carbon metabolism	5	7.30 × 10^−4^
	Arginine biosynthesis	3	5.40 × 10^−4^
	Biosynthesis of amino acids	3	3.86 × 10^−2^
	Carbon fixation in photosynthetic organisms	2	3.64 × 10^−2^
67 dpi	Metabolic pathways	56	1.46 × 10^−18^
	Biosynthesis of secondary metabolites	36	2.88 × 10^−13^
	Biosynthesis of amino acids	14	1.04 × 10^−7^
	Glycine, serine and threonine metabolism	9	1.04 × 10^−7^
	Carbon metabolism	9	1.30 × 10^−3^
	Arginine biosynthesis	7	2.83 × 10^−7^
	Ribosome	7	3.26 × 10^−2^
	Amino sugar and nucleotide sugar metabolism	6	2.80 × 10^−3^
	Alanine, aspartate and glutamate metabolism	5	4.80 × 10^−4^
	Carbon fixation in photosynthetic organisms	5	1.60 × 10^−3^
	2-Oxocarboxylic acid metabolism	5	1.80 × 10^−3^
	Glyoxylate and dicarboxylate metabolism	5	1.90 × 10^−3^
	Glycolysis/Gluconeogenesis	5	9.10 × 10^−3^
	Arginine and proline metabolism	4	4.60 × 10^−3^
	Pyruvate metabolism	4	1.92 × 10^−2^
	Nitrogen metabolism	3	2.07 × 10^−2^
	Ascorbate and aldarate metabolism	3	2.32 × 10^−2^
	Valine, leucine and isoleucine degradation	3	2.32 × 10^−2^
	Pentose phosphate pathway	3	3.41 × 10^−2^
	Proteasome	3	3.41 × 10^−2^
	Histidine metabolism	2	3.26 × 10^−2^
	Other glycan degradation	2	3.41 × 10^−2^

^a^ false discovery rate.

**Table 3 proteomes-09-00041-t003:** Differentially expressed proteins between mock-inoculated and SACMV-infected leaves at 67 dpi. Proteins with at least a ratio of 2.0 in one landrace are presented here.

Accession	Manes ID	Protein Name	Unique Peptides	q Value	T200 Ratio ^a^	TME3 Ratio ^a^
tr|A0A2C9U824	Manes.16G022000	PKS_ER domain-containing protein	3.00	2.83 × 10^−4^	2.38	−1.15
tr|A0A2C9VEA2	Manes.08G075000	Uncharacterized protein	4.00	7.56 × 10^−4^	2.37	−2.04
tr|A0A2C9U4J9	Manes.17G026300	Lactoylglutathione lyase	2.00	2.59 × 10^−3^	−1.35	2.82
tr|A0A2C9VIL5	Manes.07G049300	Proline iminopeptidase	7.00	3.91 × 10^−3^	2.26	1.05
tr|A0A0A1E5H6	Manes.11G078100	Phosphate transporter	2.00	4.63 × 10^−3^	5.62	1.01
tr|A0A251JR98	Manes.11G020100	Germin-like protein	6.00	5.29 × 10^−3^	−3.26	−1.03
tr|A0A2C9VJP2	Manes.07G087900	Glycine cleavage system H protein	2.00	6.82 × 10^−3^	−3.32	1.43
tr|A0A2C9WL85	Manes.01G155900	Uncharacterized protein	2.00	6.82 × 10^−3^	2.46	−2.07
tr|A0A2C9V1F6	Manes.11G148200	Uncharacterized protein	3.00	7.50 × 10^−3^	−2.79	5.63
tr|A0A2C9UCJ8	Manes.15G029400	Cysteine synthase	10.00	7.85 × 10^−3^	2.94	1.25
tr|A0A2C9VK05	Manes.07G093600	GLOBIN domain-containing protein	2.00	1.11 × 10^−2^	1.14	3.05
tr|A0A2C9UGM3	Manes.15G103900	Peroxidase	9.00	1.16 × 10^−2^	2.09	−1.03
tr|A0A2C9UYJ6	Manes.11G012700	Aminotran_1_2 domain-containing protein	10.00	1.16 × 10^−2^	1.01	2.01
tr|A0A2C9VAT8	Manes.09G071100	Uncharacterized protein	2.00	1.22 × 10^−2^	2.23	1.27
tr|A0A2C9VSW1	Manes.05G029200	AB hydrolase-1 domain-containing protein	2.00	1.26 × 10^−2^	2.09	1.25
tr|A0A2C9WBE7	Manes.02G064500	PALP domain-containing protein	3.00	1.30 × 10^−2^	3.27	1.14
tr|A0A2C9URN3	Manes.13G132700	Uncharacterized protein	4.00	1.32 × 10^−2^	−1.40	3.51
tr|A0A2C9VLP8	Manes.06G006200	Uncharacterized protein	3.00	1.34 × 10^−2^	3.25	−1.85
tr|A0A2C9UGX0	Manes.15G172800	Phosphoacetylglucosamine mutase	9.00	1.49 × 10^−2^	2.46	−1.12
tr|A0A2C9VP29	Manes.06G081300	Uncharacterized protein	3.00	1.51 × 10^−2^	18.46	−1.91
tr|A0A2C9UBX9	Manes.15G008100	Bet_v_1 domain-containing protein	9.00	1.76 × 10^−2^	−1.11	3.20
tr|A0A2C9V4E1	Manes.10G042700	Uncharacterized protein	2.00	1.76 × 10^−2^	−1.20	2.12
tr|A0A2C9WLB8	Manes.01G085900	Uncharacterized protein	2.00	1.76 × 10^−2^	1.17	2.43
tr|A0A251K3C9	Manes.09G021000	HP domain-containing protein	18.00	1.81 × 10^−2^	2.18	−1.71
tr|A0A2C9UTJ3	Manes.12G039500	Uncharacterized protein	6.00	1.81 × 10^−2^	−1.18	2.10
tr|A0A2C9W314	Manes.04G086800	SUMO-activating enzyme subunit	2.00	1.81 × 10^−2^	−5.15	4.65
tr|A0A251K980	Manes.08G011100	Uncharacterized protein	3.00	1.81 × 10^−2^	11.82	−2.66
tr|A0A251K915	Manes.08G008000	Uncharacterized protein	4.00	1.81 × 10^−2^	1.82	−2.13
tr|A0A2C9VGL2	Manes.08G075100	VWFA domain-containing protein	11.00	2.00 × 10^−2^	3.13	−1.29
tr|A0A2C9WEH9	Manes.02G089200	Uncharacterized protein	6.00	2.07 × 10^−2^	2.41	1.99
tr|A0A2C9UID8	Manes.15G186400	Uncharacterized protein	6.00	2.07 × 10^−2^	4.47	−5.30
tr|A0A2C9UV72	Manes.12G093300	Glycosyltransferase	2.00	2.24 × 10^−2^	9.06	−1.92
tr|A0A2C9VFU9	Manes.08G126700	Protein kinase domain-containing protein	2.00	2.66 × 10^−2^	23.94	−1.43
tr|A0A2C9W2J9	Manes.04G146100	Uncharacterized protein	7.00	2.66 × 10^−2^	2.21	−1.44
tr|A0A2C9UPJ8	Manes.13G061700	FAS1 domain-containing protein	10.00	2.66 × 10^−2^	−1.45	−2.01
tr|A0A2C9U6Y4	Manes.17G080500	Uncharacterized protein	5.00	2.67 × 10^−2^	−2.30	1.59
tr|A0A2C9U1Y7	Manes.18G103000	Uncharacterized protein	3.00	2.68 × 10^−2^	3.18	−1.22
tr|A0A2C9UQH1	Manes.13G092100	AB hydrolase-1 domain-containing protein	18.00	2.70 × 10^−2^	1.30	2.76
tr|A0A2C9UYF6	Manes.11G055100	Phospholipase D	34.00	2.73 × 10^−2^	2.05	−1.17
tr|A0A251K1M8	Manes.10G151400	Uncharacterized protein	2.00	2.73 × 10^−2^	−3.90	−1.14
tr|A0A2C9WAR9	Manes.03G202900	Uncharacterized protein	9.00	2.73 × 10^−2^	2.28	1.45
tr|A0A2C9UXW4	Manes.12G139100	Ribosomal_L7Ae domain-containing protein	3.00	2.73 × 10^−2^	−5.24	2.18
tr|A0A2C9UYT0	Manes.12G153700	Uncharacterized protein	9.00	2.79 × 10^−2^	2.40	1.20
tr|A0A2C9UNI9	Manes.14G166500	Non-specific lipid-transfer protein	6.00	2.81 × 10^−2^	−5.88	10.33
tr|A0A2C9V3Z4	Manes.10G027600	ATPase_AAA_core domain-containing protein	3.00	2.89 × 10^−2^	−52.29	6.89
tr|A0A251JSV5	Manes.11G063000	Uncharacterized protein	3.00	2.94 × 10^−2^	2.16	1.11
tr|A0A2C9WF62	Manes.02G137700	Alpha-mannosidase	13.00	3.05 × 10^−2^	2.32	1.63
tr|A0A2C9W858	Manes.03G096900	M20_dimer domain-containing protein	3.00	3.05 × 10^−2^	2.47	1.62
tr|A0A2C9VPI9	Manes.06G102100	Uncharacterized protein	2.00	3.05 × 10^−2^	2.09	−2.96
tr|A0A2C9VT96	Manes.06G155500	Glyco_hydro_18 domain-containing protein	4.00	3.05 × 10^−2^	2.67	2.71
tr|A0A2C9VIN8	Manes.07G051800	Uncharacterized protein	3.00	3.07 × 10^−2^	2.44	1.35
tr|A0A2C9UXQ1	Manes.11G033300	TauD domain-containing protein	4.00	3.14 × 10^−2^	2.87	1.67
tr|A0A0C4ZQZ2	Manes.12G135200	Annexin	13.00	3.14 × 10^−2^	2.78	1.41
tr|A0A2C9UCR5	Manes.15G007900	Bet_v_1 domain-containing protein	2.00	3.14 × 10^−2^	−1.08	3.69
tr|A0A2C9U3Q4	Manes.17G010500	RRM domain-containing protein	4.00	3.14 × 10^−2^	2.41	−1.45
tr|A0A2C9WMF8	Manes.01G202400	Uncharacterized protein	3.00	3.21 × 10^−2^	6.53	−1.13
tr|A0A2C9VRV6	Manes.06G177500	Uncharacterized protein	2.00	3.26 × 10^−2^	3.26	1.27
tr|A0A2C9UUG2	Manes.12G027700	Uncharacterized protein	14.00	3.32 × 10^−2^	2.28	1.56
tr|A0A2C9U185	Manes.18G062000	NAD(P)H-hydrate epimerase	7.00	3.42 × 10^−2^	4.79	−2.12
tr|A0A2C9V3P1	Manes.10G055500	WD_REPEATS_REGION domain-containing protein	2.00	3.43 × 10^−2^	2.24	−1.84
tr|A0A2C9VBZ8	Manes.09G142200	Uncharacterized protein	2.00	3.59 × 10^−2^	−1.63	67.74
tr|A0A2C9UV10	Manes.12G095700	AB hydrolase-1 domain-containing protein	3.00	3.59 × 10^−2^	1.42	2.21
tr|A0A2C9UJE4	Manes.14G034500	UMP-CMP kinase	2.00	3.59 × 10^−2^	3.10	1.14
tr|A0A2C9WRD2	Manes.01G245400	Importin N-terminal domain-containing protein	9.00	3.59 × 10^−2^	3.16	−1.16
tr|A0A251KS16	Manes.05G011800	Uncharacterized protein	12.00	3.61 × 10^−2^	2.05	−1.21
tr|A0A2C9UGV6	Manes.15G140500	Eukaryotic translation initiation factor 3 subunit I	10.00	3.61 × 10^−2^	2.10	−1.40
tr|A0A2C9UA86	Manes.16G087600	3-ketoacyl-CoA synthase	2.00	3.61 × 10^−2^	8.18	−6.12
tr|A0A2C9UX42	Manes.12G156900	Hist_deacetyl domain-containing protein	5.00	3.74 × 10^−2^	−4.08	1.33
tr|A0A2C9W735	Manes.03G081200	FAD-binding PCMH-type domain-containing protein	8.00	3.77 × 10^−2^	5.66	−1.54
tr|A0A2C9VMD1	Manes.07G122600	AAI domain-containing protein	4.00	3.77 × 10^−2^	−10.20	−1.40
tr|A0A2C9VAA7	Manes.09G127700	Bet_v_1 domain-containing protein	7.00	3.77 × 10^−2^	1.32	2.22
tr|A0A2C9URF0	Manes.13G126100	Uncharacterized protein	2.00	3.77 × 10^−2^	2.16	−1.04
tr|A0A2C9U7U7	Manes.16G007800	Uncharacterized protein	7.00	3.77 × 10^−2^	−2.85	1.88
tr|A0A2C9W3J5	Manes.04G095600	eRF1_1 domain-containing protein	4.00	3.77 × 10^−2^	2.26	−1.02
tr|A0A199UC10	Manes.s021400	ZnMc domain-containing protein	2.00	3.77 × 10^−2^	3.57	1.09
tr|A0A2C9WH55	Manes.01G012700	Vac14_Fab1_bd domain-containing protein	3.00	3.80 × 10^−2^	2.00	1.10
tr|A0A2C9U3Q3	Manes.18G143800	Uncharacterized protein	3.00	3.80 × 10^−2^	−5.26	1.18
tr|A0A2C9V4F7	Manes.10G077300	Uncharacterized protein	2.00	3.80 × 10^−2^	2.46	1.12
tr|A0A251LVH3	Manes.01G267300	Uncharacterized protein	3.00	3.87 × 10^−2^	1.54	2.11
tr|A0A2C9UJT8	Manes.14G080800	Peptidyl-prolyl cis-trans isomerase	2.00	3.90 × 10^−2^	−3.45	4.03
tr|A0A2C9V4B1	Manes.10G077100	Lipase_3 domain-containing protein	5.00	3.92 × 10^−2^	2.03	−1.24
tr|A0A2C9VG07	Manes.08G081300	Uncharacterized protein	2.00	3.93 × 10^−2^	−5.02	6.17
tr|A0A2C9WEB2	Manes.02G155900	Uncharacterized protein	4.00	3.93 × 10^−2^	1.17	4.80
tr|A0A2C9V5Y8	Manes.10G128500	Uncharacterized protein	5.00	3.93 × 10^−2^	4.19	−1.96
tr|A0A2C9WJN9	Manes.01G112700	AAI domain-containing protein	3.00	4.06 × 10^−2^	−2.73	6.45
tr|A0A2C9VN74	Manes.06G056000	Uncharacterized protein	2.00	4.06 × 10^−2^	3.26	−1.66
tr|A0A2C9WNU1	Manes.01G247800	Glycosyltransferase	2.00	4.06 × 10^−2^	2.17	1.39
tr|A0A2C9VYT5	Manes.05G155000	AAA domain-containing protein	9.00	4.06 × 10^−2^	2.82	−1.07
tr|A0A2C9W393	Manes.04G165200	Uncharacterized protein	4.00	4.06 × 10^−2^	4.42	−1.42
tr|A0A2C9VKB0	Manes.07G107200	Chalcone-flavonone isomerase family protein	10.00	4.06 × 10^−2^	2.89	1.20
tr|A0A2C9UBT3	Manes.15G008400	Bet_v_1 domain-containing protein	7.00	4.06 × 10^−2^	2.85	13.10
tr|A0A2C9V9W8	Manes.09G113900	Uncharacterized protein	4.00	4.06 × 10^−2^	2.31	1.48
tr|A0A2C9WD78	Manes.02G122500	Alpha-1,4 glucan phosphorylase	5.00	4.06 × 10^−2^	2.32	4.83
tr|A0A2C9WCC9	Manes.02G095600	CTP_transf_like domain-containing protein	3.00	4.18 × 10^−2^	1.16	2.35
tr|A0A2C9UNS6	Manes.13G043200	AAA domain-containing protein	9.00	4.22 × 10^−2^	4.75	−1.32
tr|A0A2C9U6K5	Manes.17G064800	Uncharacterized protein	2.00	4.23 × 10^−2^	−4.61	1.83
tr|A0A2C9VE34	Manes.08G069100	Uncharacterized protein	15.00	4.30 × 10^−2^	2.06	1.34
tr|A0A2C9V5M6	Manes.10G121900	Uncharacterized protein	13.00	4.30 × 10^−2^	2.52	−1.27
tr|A0A2C9U8P4	Manes.17G111900	Uncharacterized protein	3.00	4.30 × 10^−2^	2.33	1.50
tr|A0A2C9USP9	Manes.12G024300	Uncharacterized protein	2.00	4.30 × 10^−2^	2.64	−1.54
tr|A0A2C9V2I1	Manes.10G019200	Uncharacterized protein	2.00	4.34 × 10^−2^	3.30	−1.16
tr|A0A2C9VWB9	Manes.05G146600	Peptidyl-prolyl cis-trans isomeras	3.00	4.34 × 10^−2^	−1.52	2.06
tr|A0A2C9UGS7	Manes.15G108300	Uncharacterized protein	6.00	4.45 × 10^−2^	3.10	−1.11
tr|A0A2C9VJ85	Manes.07G017100	60S ribosomal protein L13	2.00	4.45 × 10^−2^	−3.86	2.11
tr|A0A2C9UTD1	Manes.12G033700	Dolichyl-diphosphooligosaccharide--protein glycosyltransferase subunit 2	4.00	4.49 × 10^−2^	5.39	−1.86
tr|A0A2C9WHE0	Manes.01G032500	PCI domain-containing protein	2.00	4.52 × 10^−2^	7.14	−1.40
tr|A0A2C9UFY6	Manes.15G139800	Transmembrane 9 superfamily member	4.00	4.55 × 10^−2^	4.51	−2.11
tr|A0A2C9VXS5	Manes.05G194900	Protein kinase domain-containing protein	2.00	4.55 × 10^−2^	−8.12	−2.18
tr|A0A2C9VQH2	Manes.06G055700	Uncharacterized protein	7.00	4.57 × 10^−2^	2.10	−2.39
tr|A0A2C9VYM5	Manes.05G142300	Methyltransferase	2.00	4.62 × 10^−2^	3.32	−6.67
tr|A0A2C9TZI5	Manes.18G001100	Pyrophosphate--fructose 6-phosphate 1-phosphotransferase subunit alpha	3.00	4.62 × 10^−2^	2.13	1.34
tr|A0A2C9V233	Manes.11G128200	Uncharacterized protein	3.00	4.65 × 10^−2^	−6.76	9.16
tr|A0A2C9V6N1	Manes.09G008000	F420_oxidored domain-containing protein	4.00	4.68 × 10^−2^	1.08	2.91
tr|A0A251JLF7	Manes.12G028400	Carbonic anhydrase	2.00	4.68 × 10^−2^	2.95	−1.11
tr|A0A2C9W4R1	Manes.03G053100	Uncharacterized protein	5.00	4.68 × 10^−2^	−5.50	1.68
tr|A0A2C9V313	Manes.10G032500	NAD(P)-bd_dom domain-containing protein	3.00	4.68 × 10^−2^	3.11	1.28
tr|A0A2C9VBY7	Manes.09G183600	Uncharacterized protein	3.00	4.68 × 10^−2^	3.66	−1.04
tr|A0A2C9UMZ5	Manes.13G011400	AAI domain-containing protein	5.00	4.68 × 10^−2^	−9.13	6.96
tr|A0A2C9VRT4	Manes.06G097000	Plasma membrane ATPase	6.00	4.69 × 10^−2^	5.75	−1.45
tr|A0A2C9WA59	Manes.03G162900	Ribosomal_L14e domain-containing protein	2.00	4.69 × 10^−2^	4.36	−1.86
tr|A0A2C9VYM8	Manes.04G011900	Carboxypeptidase	3.00	4.69 × 10^−2^	−1.36	3.72
tr|A0A251K5T0	Manes.09G086600	Uncharacterized protein	6.00	4.69 × 10^−2^	3.00	−1.22
tr|A0A2C9V462	Manes.10G078000	Glycosyltransferase	5.00	4.71 × 10^−2^	3.12	−1.63
tr|A0A2C9VLD9	Manes.07G091600	Uncharacterized protein	4.00	4.81 × 10^−2^	4.30	−2.64
tr|A0A2C9W8C0	Manes.03G123300	Trafficking protein particle complex subunit	3.00	4.81 × 10^−2^	2.54	−1.12
tr|A0A2C9W8D3	Manes.03G089200	Uncharacterized protein	2.00	4.87 × 10^−2^	−14.91	2.30
tr|A0A199U962	Manes.S101100	Lipase_GDSL domain-containing protein	3.00	4.87 × 10^−2^	−2.51	3.28
tr|A0A2C9UL26	Manes.14G124000	Protein kinase domain-containing protein	3.00	4.89E × 10^−2^	−1.70	−2.19
tr|A0A2C9U8J5	Manes.16G001400	Abhydrolase_2 domain-containing protein	2.00	4.89 × 10^−2^	−1.41	8.09
tr|A0A2C9VH42	Manes.08G171700	Non-specific lipid-transfer protein	3.00	4.97 × 10^−2^	−13.04	6.81
tr|A0A2C9VWH1	Manes.05G151600	Uncharacterized protein	10.00	4.97 × 10^−2^	3.19	−3.49

^a^ Negative ratio = under-expression, positive ratio = over-expression.

## Data Availability

The datasets generated and analysed during the current study are available from the corresponding author on request.

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
