# Peer review of "Proteome Mapping of South African Cassava Mosaic Virus-Infected Susceptible and Tolerant Landraces of Cassava"

_proteomes, 2021, doi:10.3390/proteomes9040041_

Round 1

Reviewer 1 Report

Authors undertake the important task to report the proteome response to SACMV infection in casava plants with the aim to determine the mechanisms that is responsible for increased resistance of the immune response in landrace TME3. This work is important and need to be published, but require the following corrections/attention:

  1. CMD Symptoms Scoring Scale – the method is based on subjective observation and represent quasi-quantification which might not support statistical analysis. Therefore, I think that graph b in fig 1 should just show the difference in avg. score (no statistical significance of the result). Otherwise, maybe the analysis of chlorophyl content or leaf shape/convex area/size (scanning and imageJ analysis) would be better for quantification of CMD and could be subjected to statistical analysis. Nevertheless the methods section must include precise description of the scale used, including the border condition e.g. how authors differentiate between mild, slight, severe, strong - deformation, chlorosis, mosaic (e.g. at which point authors name it chlorosis and when clear mosaic)... Maybe authors could estimate those vague descriptions with % e.g. of leaf with chlorosis or leaf deformation/shape change in comparison to mock.
  2. Most of the manuscript it well written but sometimes there are obvious errors, that authors should carefully correct: e.g. line 301 …gram showed differences between leave leaf proteome at 32 dpi and 67 dpi…  
  3. The whole 3.3 GO analysis part must be rewritten or redone:
    1. Authors present graph only for 67dpi for TME3 and rest of the result are literal description of SI figures. This is too long, confusing and hard to understand – I do not see the point – also those results are only in part used in discussion. Authors should group GO analysis results and decide what is important and relevant result in context of SACMV (as they use in the discussion) and report them as main result, refereeing to rest in SI. I am also confused what those results are(?), since authors describe Fig 3 as enrichment analysis, but this looks like standard GO analysis(that is % of DEP not a enrichment analysis where there is % of DEP in total protein database in each category, or frequency in analysis to frequency in total protein database/background). The enrichment analysis (e.g. GO using Panthers classification system at geneonthology.org) have to generate the background frequency, the sample frequency, expected p-value, an indication of over/underrepresentation for each term, and p-value. In short, I think this part would benefit if authors would present it as the over or under- representation of categories in most relevant comparison e.g. between landraces or up or down regulated proteins. Then it would be also clear if the chloroplast localization (mentioned in discussion) is overrepresented in TME3?
    2. Please include section in methods that describe all the GO, KEGG pathway, STRING analysis parameters.
  4. Additionally, in the paragraph 3.4 (and any other part of result) authors should add some meaningful conclusions and relate results to the aim of the study – this should be as a preface to discussion. Without a context, the result section is hard to comprehend. Authors need to concurrently report the results and their importance toward achieving the aim of the study. E.g. are the represented pathways related to plant SACMV response? Is STRING analysis show relevant pathways and connections?

I like the discussion, but it is a little bit too long, authors should look for ways to shorten it and focus on most important results and conclusions of their research and briefly refer to result that are inconclusive or reported before - expected based on other research.

Author Response

Response

Thank you to this reviewer for the time taken. Much appreciated to make this a better manuscript.

Reviewer # 1

Authors undertake the important task to report the proteome response to SACMV infection in cassava plants with the aim to determine the mechanisms that is responsible for increased resistance of the immune response in landrace TME3. This work is important and need to be published, but require the following corrections/attention:

  1. CMD Symptoms Scoring Scale – the method is based on subjective observation and represent quasi-quantification which might not support statistical analysis. Therefore, I think that graph b in fig 1 should just show the difference in avg. score (no statistical significance of the result). Otherwise, maybe the analysis of chlorophyll content or leaf shape/convex area/size (scanning and imageJ analysis) would be better for quantification of CMD and could be subjected to statistical analysis. Nevertheless the methods section must include precise description of the scale used, including the border condition e.g. how authors differentiate between mild, slight, severe, strong - deformation, chlorosis, mosaic (e.g. at which point authors name it chlorosis and when clear mosaic)... Maybe authors could estimate those vague descriptions with % e.g. of leaf with chlorosis or leaf deformation/shape change in comparison to mock.

Authors response: We have included the scoring approach in methods.

The symptom score scales of 0-5 (Fauquet and Fargette, 1990) and 1-5 (Hahn et al., 1980) have been used for CMD symptom scoring in the cassava community based on several publications, including Sseruwagi et al., 2004; Allie et al., 2014). The authors herein have used the 0-5 scale. We have explained this in methods.

We agree we should not use statistical differences and we have removed this from Figure 1. 

  1. Most of the manuscript it well written but sometimes there are obvious errors that authors should carefully correct: e.g. line 301 …gram showed differences between leaveleafproteome at 32 dpi and 67 dpi…  

Authors response: Minor errors are being re-checked and corrected.

3. The whole 3.3 GO analysis part must be rewritten or redone:

A. Authors present graph only for 67 dpi for TME3 and rest of the result are literal description of Supplementary S1 figures. This is too long, confusing and hard to understand – I do not see the point – also those results are only in part used in discussion. Authors should group GO analysis results and decide what is important and relevant result in context of SACMV (as they use in the discussion) and report them as main result, referring to rest in S1.

 Authors response: The detailed results in the GO have been removed and referred to in Supplementary S1 figures. As suggested by this reviewer, since the focus of this manuscript is recovery and SACMV tolerance in TME3, the focus for the results (and discussion) will be focused on TME3 and T200 at 67 dpi. We are also focusing on GOs with a percentage over 30% representation and have changed Figure 3 to include only those categories and added T200 from 67 dpi.

B. I am also confused what those results are (?), since authors describe Fig 3 as enrichment analysis, but this looks like standard GO analysis (that is % of DEP not a enrichment analysis where there is % of DEP in total protein database in each category, or frequency in analysis to frequency in total protein database/background). The enrichment analysis (e.g. GO using Panthers classification system at gene onthology.org) have to generate the background frequency, the sample frequency, expected p-value, an indication of over/underrepresentation for each term, and p-value. In short, I think this part would benefit if authors would present it as the over or under- representation of categories in most relevant comparison e.g. between landraces or up or down regulated proteins. Then it would be also clear if the chloroplast localization (mentioned in discussion) is over-represented in TME3?

Authors response: This comment is absolutely correct. Figure 3 is the percentage (over-representation) of genes in each of these categories. We have corrected this and the Figure legend and made this clear in the text.

C. Please include section in methods that describe all the GO, KEGG pathway, STRING analysis parameters.

Authors response: This has been done.

D. Additionally, in the paragraph 3.4 (and any other part of result) authors should add some meaningful conclusions and relate results to the aim of the study – this should be as a preface to discussion. Without a context, the result section is hard to comprehend. Authors need to concurrently report the results and their importance toward achieving the aim of the study. E.g. are the represented pathways related to plant SACMV response? Is STRING analysis show relevant pathways and connections?

Author Responses: For this section the authors will make it clear that the Kegg pathways or STRING connections relate to a host response to SACMVs and their importance. All the pathways and STRING represent the networks and interaction with differentially expressed proteins in response to SACMV and therefore are important. So it is not just the DE protein but its networks. That is why we have included them. But will make this clear.

I like the discussion, but it is a little bit too long, authors should look for ways to shorten it and focus on most important results and conclusions of their research and briefly refer to result that are inconclusive or reported before - expected based on other research.

Authors responses: The authors have already focused on the most important findings in relation to other geminiviruses and recovery in TME3 at 67 dpi but will shorten this. Conclusions will be edited to reflect the most important results.

Reviewer 2 Report

The manuscript described in detail the proteome variation of cassava following SACMV infection of a susceptible variety and a tolerant one. Results are well analyzed and permit to describe different pathways involved in Cassava tolerance/resistance to geminivirus. 

Maybe replacing gene ontology analysis results description and figure 3 by a table with the 8 features in colon (variety, dpi, protein expression) will facilitate results reading and comparativeness.

Author Response

The manuscript described in detail the proteome variation of cassava following SACMV infection of a susceptible variety and a tolerant one. Results are well analyzed and permit to describe different pathways involved in Cassava tolerance/resistance to geminivirus. 

Reviewer 2 Comments

Maybe replacing gene ontology analysis results description and figure 3 by a table with the 8 features in colon (variety, dpi, protein expression) will facilitate results reading and comparativeness.

Authors responses: In response to the comments here and the first reviewer we will reduce the GO section results description and focus mainly on T200 and TME3 at 67 dpi as this manuscript is examining features of CMD recovery at 67 dpi..  

For the main text figure for GO we will redraw the graphs with only GO of (>30%) for both T200 and TME3 at 67 dpi (Up-regulated and down-regulated separately) and then the remaining GO (<30%) will be summarised in a Table that will form part of the supplementary data.

Reviewer 3 Report

In the manuscript, Ramulifho & Rey identified the cassava leaf proteome involved in defense against the South African cassava mosaic virus. This project is interesting, and I believe the article would provide helpful information for controlling the devastating virus. However, the abstract, results and discussion might need extra editing from my modest point of view. The article is redundant, and some sentences are unnecessary. Please find the suggestions in detail, as below.
Major concerns:
The title of the manuscript is not clear. A title like “Proteome mapping of South African cassava mosaic virus-infected susceptible and tolerant landraces of cassava” might be more suitable.
For the abstract, some sentences do not describe the major data in the manuscript. Sentences “Ribosomal_L7Ae domain-containing protein ….. in TME3 (line 21-24)” should be in the discussion.
I don’t think the conclusion in the abstract reflects the content of the manuscript. "The proteomic differences that need further investigation" cannot be a conclusion. I suggest the authors redraft this and even the section “Conclusions”
Fig 1a should be rearranged. The quality of the presentation is way from publication.
In the section “Gene Ontology analysis”, is there a way to represent the data more clearly? Please consider using a pie chart to distinguish the number of differentially expressed proteins and a table to present the pathways where these proteins belong.
The resolution of Figure 4 needs to be improved.
I was confused by the aim of the section “Validation of changes in RNA level”. Do the authors get some positive results to support the data collected from the proteomic analysis?
I suggest the authors concise the discussion. It is long and redundant. Please select several key genes for discussion, especially those tested by RT-qPCR.

Minor concerns:
Line 18, “chloroplast, proteasome, and ribosome” should be “chloroplast, proteasome, and ribosome”
Line 292, “Ten proteins were similar to both TME3 and T200 at both timepoints”. Please clarify the sense.
Table 1 title, “SACMV infected” should be “SACMV-infected”.
Line 319, 325, delete “response to”

Author Response

Reviewer # 3
In the manuscript, Ramulifho & Rey identified the cassava leaf proteome involved in defense against the South African cassava mosaic virus. This project is interesting, and I believe the article would provide helpful information for controlling the devastating virus. However, the abstract, results and discussion might need extra editing from my modest point of view. Please find the suggestions in detail, as below.

Major concerns:
1. The title of the manuscript is not clear. A title like “Proteome mapping of South African cassava mosaic virus-infected susceptible and tolerant landraces of cassava” might be more suitable.
Authors response: The title has been changed

2A. For the abstract, some sentences do not describe the major data in the manuscript.

Authors response: In an abstract of 205 words, to explain the manuscript in brief and give more major data conclusions in relation to GO/Kegg/protein-interaction results is not feasible. The three overall conclusions/results are given in the abstract (i) that the number of DE proteins between T200 and TME3 differ; (ii) expression profiles differ between the landraces and time points (iii) defence-associated pathways such as the chloroplast, proteasome and ribosome are over-represented at 67 dpi in SACMV-tolerant TME3 (which is the focus of the paper). It is not possible to add more conclusions as there is a lot of data and the abstract length is restricted. So we then chose to add what we consider the  most significant result in terms of individual proteins and geminiviruses that will be of interest to our target audience. In fact this is only the second report of RPL10 as a player in the NIK1-mediated effector triggered immunity (ETI) response to geminivirus infection (Fontes et al are the other group). And first report in cassava for CMD. So highly significant. We cannot add anything more.

2B. I don’t think the conclusion in the abstract reflects the content of the manuscript. "The proteomic differences that need further investigation" cannot be a conclusion.

Authors responses: We agree and have removed this. This has been replaced with “In conclusion, differential protein expression responses in TME3 and T200 may be key to unravelling tolerance to CMD.

2C I suggest the authors redraft “Conclusions”
Authors response:  We will do this in light of the comments.

  1. Fig 1a should be rearranged.
    Authors response:  We have responded to the comment about Figure 1 from reviewer 1 and have edited this according to those comments. Not sure what other rearrangements are required?
  1. In the section “Gene Ontology analysis”, is there a way to represent the data more clearly?

Please consider using a pie chart to distinguish the number of differentially expressed proteins and a table to present the pathways where these proteins belong.
Authors response:

The resolution of Figure 4 needs to be improved.

Authors response: The only way to improve the quality here is to separate the two networks into two separate figures and enlarge those figures. We are doing this.

I was confused by the aim of the section “Validation of changes in RNA level”. Do the authors get some positive results to support the data collected from the proteomic analysis? Response:  Yes.

I suggest the authors concise the discussion. It is long and redundant. Please select several key genes for discussion, especially those tested by RT-qPCR.

Authors response: The discussion will be edited according this comment and the other reviewers. We will cut it down and focus on key proteins and especially those at 67 dpi.  We have selected proteins associated with other geminivirus and plant virus studies from the literature that have been shown to be significant and of interest to the cassava and geminivirus community. Conclusions will be edited around this reviewer comment and those of reviewer 1 which was the same.

Minor concerns:

Author responses: These below will be corrected.

Line 18, “chloroplast, proteasome, and ribosome” should be “chloroplast, proteasome, and ribosome”
Line 292, “Ten proteins were similar to both TME3 and T200 at both timepoints”. Please clarify the sense.
Table 1 title, “SACMV infected” should be “SACMV-infected”.
Line 319, 325, delete “response to”

Round 2

Reviewer 3 Report

The authors have made some modifications to the manuscript. Some of the suggestions/comments were addressed in the revised manuscript. However, for some reason, I did not get all the point-to-point responses in the rebuttal letter. Below, please find my clarification for some of the comments.

“Fig 1a should be rearranged”

I mean the quality/presentation of Figure 1 does not match the requirement for publication. At least the authors should align the images by cropping/enlarging individual ones. In addition, the Figure 1b and 1c are all charts, thus, they both should have the background horizontal lines.

“Validation of changes in RNA level”

“Validation” can only be used when the authors confirmed the changes of a specific protein/gene in RNA level and protein level. If the changes in the protein level and the RNA level are opposite, the word “Validation” is not suitable. Refer to the case as stated in the manuscript “Transcripts of receptor-like kinases (Manes.05G194900, Manes.14G124000), Kinase 14 (Manes.08G126700) and receptor-like protein (Manes.06G055700) were over-expressed with a fold change of 7.9, 3.8, 1.7 and 2.5, respectively (Figure 5), whereas, the expression levels of their proteins were under-expressed based on the proteomics results (Table 3).”

Even with separation, Figure 4 is difficult to read. The node notes are quite blurred. There should be ways to improve the resolution.

Line 93-94. Please provide the full name of these genes first. The abbreviation should be italicized.

Line 25, “Ravelling” should be “ravel”.

Though I believe the significance of this work is high, the resolutions of some figures need extra edits. Minor language (grammar/ punctuation) mistakes were found. 

Author Response

Responses to Reviewer 1 (2)

The authors have made some modifications to the manuscript. Some of the suggestions/comments were addressed in the revised manuscript. However, for some reason, I did not get all the point-to-point responses in the rebuttal letter. Below, please find my clarification for some of the comments.

“Fig 1a should be rearranged”

I mean the quality/presentation of Figure 1 does not match the requirement for publication. At least the authors should align the images by cropping/enlarging individual ones. In addition, the Figure 1b and 1c are all charts, thus, they both should have the background horizontal lines.

Author responses:   Re-arrangement and enlargement/cropping will be done. Figure 1c has been corrected.

“Validation of changes in RNA level”

“Validation” can only be used when the authors confirmed the changes of a specific protein/gene in RNA level and protein level. If the changes in the protein level and the RNA level are opposite, the word “Validation” is not suitable. Refer to the case as stated in the manuscript “Transcripts of receptor-like kinases (Manes.05G194900, Manes.14G124000), Kinase 14 (Manes.08G126700) and receptor-like protein (Manes.06G055700) were over-expressed with a fold change of 7.9, 3.8, 1.7 and 2.5, respectively (Figure 5), whereas, the expression levels of their proteins were under-expressed based on the proteomics results (Table 3).”

Authors response: We have changed the title to:  “Changes in RNA expression levels"

Even with separation, Figure 4 is difficult to read. The node notes are quite blurred. There should be ways to improve the resolution.

Authors response:  Figure 4 has been re-downloaded as "high resolution bit map" and this has improved the quality.

Line 93-94. Please provide the full name of these genes first. The abbreviation should be italicized. 

Line 25, “Ravelling” should be “ravel”.

Authors response: These have both been corrected.

Though I believe the significance of this work is high, the resolutions of some figures need extra edits. Minor language (grammar/ punctuation) mistakes were found. 

Authors response: Minor errors have been corrected where found. We have improved resolution and clarity of all the STRING figures.  
